# Detection of m6A from direct RNA sequencing using a multiple instance learning framework

Christopher Hendra[1,2,3], Ploy N. Pratanwanich[2,4,5], Yuk Kei Wan[2,6], W. S. Sho Goh [7], Alexandre Thiery [3] ✉ & Jonathan Göke [2,3,8] ✉

RNA modifications such as m6A methylation form an additional layer of complexity in the transcriptome. Nanopore direct RNA sequencing can capture this information in the raw current signal for each RNA molecule, enabling the detection of RNA modifications using supervised machine learning. However, experimental approaches provide only site-level training data, whereas the modification status for each single RNA molecule is missing. Here we present m6Anet, a neural-network-based method that leverages the multiple instance learning framework to specifically handle missing read-level modification labels in site-level training data. m6Anet outperforms existing computational methods, shows similar accuracy as experimental approaches, and generalizes with high accuracy to different cell lines and species without retraining model parameters. In addition, we demonstrate that m6Anet captures the underlying read-level stoichiometry, which can be used to approximate differences in modification rates. Overall, m6Anet offers a tool to capture the transcriptome-wide identification and quantification of m6A from a single run of direct RNA sequencing.

Modifications in RNA nucleotides were first discovered in the 1950s[1,2], and today, more than 150 different modifications have been described[3,4]. One of the most common RNA modifications is m6A, the main internal methylation on mammalian mRNA[5,6]. This modification presents mostly at the consensus motif DRACH (D–A, G, or U, R–A or G while H is A, C or U) and has been shown to impact RNA structure[7], stability[8,9], splicing[10], and translation[11]. Disruption of m6A homeostasis in animal models affects regulation of stem cells[12,13], fertility and the developmental process[14], while in humans, this modification plays an important role in cancer[15,16], cell-fate transition and determination[17,18] and transition, development[19], and diseases[20,21].

 Experimental identification of trasncriptome-wide RNA modifications can be achieved with three main approaches.

Immunoprecipitation methods such as MeRIP-Seq[22], m6A-Seq[23], PA-m6A-Seq[24], m6A-CLIP/IP[25], miCLIP[26], m6A-LAIC-Seq[27], m6ACE-Seq[28], and m6A-Seq2[29] use antibodies that specifically bind to the modified ribonucleotide. Chemical-based detection methods such as Pseudo-Seq[30], AlkAniline-Seq[31], utilize chemical compounds that selectively react with the modified ribonucleotide. Approaches such as Mazter-Seq[32], m6A-REF-Seq[33], or DART-Seq[34] use specific enzymes to selectively distinguish modified and unmodified bases. These approaches are similar in that they isolate the RNA after inducing changes to the surrounding nucleotides, followed by reverse transcription and short-read cDNA sequencing to detect these changes. While these approaches provide a transcriptome-wide map of RNA-modification sites, they are limited by the availability of antibodies or compounds for specific

[1]Institute of Data Science, National University of Singapore, Singapore, Singapore. [2]Genome Institute of Singapore, A*STAR, Singapore, Singapore. [3]Department of Statistics and Data Science, National University of Singapore, Singapore, Singapore. [4]Department of Mathematics and Computer Science, Faculty of Science, Chulalongkorn University, Chulalongkorn, Thailand. [5]Chula Intelligent and Complex Systems Research Unit, Chulalongkorn University, Chulalongkorn, Thailand. [6] Yong Loo Lin School of Medicine, National University of Singapore, Singapore, Singapore. [7]Institute of Molecular Physiology, Shenzhen Bay Laboratory, Shenzhen, China. [8]National Cancer Center of Singapore, Singapore, Singapore. ✉e-mail: a.h.thiery@nus.edu.sg; gokej@gis.a-star.edu.sg

modifications[35]. Also they lack single-nucleotide resolution[22,23] and are incapable of identifying modifications for single RNA molecules.

The ability to sequence native RNA using Nanopore direct RNA sequencing (RNA-Seq) can potentially overcome these limitations[36]. Direct RNA-Seq infers the RNA sequence using the current intensity when RNA molecules pass through the pores. Modified nucleotides will emit a different signal intensity compared to unmodified nucleotides, allowing the computational identification of modified sites for each individual RNA molecule using either supervised or comparative approaches[37]. Comparative approaches do not require training data for known RNA modifications but instead use control or reference samples to detect meaningful shifts in signal-based features that correlate to the presence of modifications. Comparative methods such as Tombo[38], DRUMMER[39], nanoDOC[40], Nanocompore[41], ELIGOS[42], xPore[43], and Yanocomp[44] detect m6A sites by comparing against samples without m6A modifications. While these methods are accurate, their success relies on the availability of m6A-free control samples. This involves silencing of specific writer genes, which can be a limiting factor.

Supervised detection of m6A modifications involves training a classifier using labels that can be obtained from synthetically modified RNA samples or experimental protocols such as miCLIP[26], MeRIP-Seq[22] or m6ACE-Seq[28]. Methods such as EpiNano[45,46], MINES[47], and nanom6A[48], use training data to identify m6A using the sequencing error profile or shifts in the current signal intensity. Supervised methods can be applied on a single sample, overcoming the main limitation of comparative methods for detection of specific RNA modifications. Furthermore, given the availability of training data, one can retrain supervised classifiers to detect other modifications such as pseudouridine, which is detected by NanoRMS[49]. However, existing approaches are limited to a specific nucleotide content[45–48], and they are currently less accurate than comparative approaches using an m6A-free control[41,43,44].

One of the main challenges for supervised approaches is that training labels are provided for a set of reads at the site level, but not for each individual read, which is known as a multiple instance learning (MIL) problem[50,51]. Existing methods address this problem by averaging read-based features[45–47]. However, at any given site, we are likely to have a mixture of modified and unmodified reads and as such, not all reads provide useful features to detect m6A sites. Therefore, current approaches, which do not consider the MIL structure in the training data might fail to detect m6A modifications from sites with low stoichiometry as it tends to obscure signals from the lowly expressed modified RNAs, and it limits the ability to integrate variation in read-level features into a predictive model.

To address these limitations, we developed *m6Anet*, a MIL-based neural network model that takes in signal intensity and sequence features to identify potential m6A sites from direct RNA-Seq data. Our model takes into account the mixture of modified and unmodified RNAs and outputs the m6A-modification probability at any given site for all DRACH fivemers represented in the training data. Unlike existing approaches, m6Anet learns a high-dimensional representation of individual reads from each candidate site before aggregating them together to produce a more accurate prediction of m6A sites. By applying m6Anet to human, *Arabidopsis*, and synthetic direct RNA-Seq data we demonstrate that it detects previously unlabeled m6A sites and generalizes across different cell lines and species without a reduction in performance. The approach is general enough that the network can be retrained to classify any natural or artificial RNA modifications given a set of labels.

## Results

### m6ANet identifies methylated positions with a MIL approach

Here we present m6Anet, a neural-network-based MIL model that combines learning the representation of each individual read with classifying m6A-modified sites. m6Anet comprises two separate modules that

are optimized jointly—a read-level encoder and a pooling layer. The read-level encoder uses signal and sequence features from each read, and transforms them into a high-dimensional representation before predicting the probability of each read being modified (Fig. 1a). The read-level probability is then pooled to give a probability estimate that a site is modified (Fig. 1a). By combining features that represent signal and sequence properties, m6Anet can learn a model that can be applied for all fivemers that are represented in the training data. Furthermore, the end-to-end training of our model implicitly learns a representation of the data that is optimized towards predicting the probability that a site is modified on the basis of the assumption encoded within the pooling layer. In our case, the pooling layer represents the probability that a particular site contains at least one modified position, but in practice one can choose a pooling layer that best captures the labeling process associated with the data collection step. While we apply m6Anet to the task of m6A RNA-modification detection, the framework can also be applied to other tasks for which training labels are available, such as detection of DNA modification or other RNA modifications of interest. m6Anet is implemented in Python and available from GitHub (https://github.com/GoekeLab/m6anet).

### Training data for m6Anet model parameter estimation

To learn the model parameters, m6Anet requires training data consisting of labels (modified/unmodified) and direct RNA-Seq reads. To train a model for m6A we used labels obtained from m6ACE-Seq that identifies m6A at single-nucleotide resolution[28]. m6Anet uses positions that are identified to have m6A as labels for the 'modified' class, and any other position with the same fivemer sequences that are included in the modified class will be used as the 'unmodified' class. To extract features for model training and predictions, we used nanopolish[52] eventalign to segment nanopore raw signals into their respective positions in the transcriptome. Since m6A modifications occur at the DRACH motifs, we removed any non-DRACH motifs from these data for m6Anet; however, this step is not required for training data without prior knowledge about the motifs. Since m6A modifications are rare compared to unmodified sites, we oversample the modified sites during training to obtain a balanced dataset (see Supplementary Text for results with alternative sampling strategies). Here we used direct RNA-Seq data from the HCT116 cell line for which matched m6ACE-Seq data is available as part of the Singapore Nanopore Expression Project[53].

### Contribution of signal and sequence features

m6Anet uses signal features corresponding to the normalized signal intensity, standard deviation, and dwelling time for each position. To understand how each feature contributes to the prediction of m6Anet, we explored the difference in features between the predicted modified and unmodified sites for each of the DRACH motifs. Signal intensity of the center base showed the strongest difference between predicted modified and unmodified sites, with dwell time showing the smallest difference (Fig. 1b and Supplementary Fig. 1a–r). Overall, all features distinguish modified and unmodified sites and are informative for m6A predictions.

As RNA modifications can affect the nanopore current signal at the neighboring bases, we tested whether information from additional positions increases the model accuracy. We performed fivefold cross validation with features extracted from 0 to 5 base pairs flanking the candidate sites to evaluate the additional value of neighboring positions, splitting the data at the gene level to ensure independence between training and test set. Our results show that m6Anet performance is highest when one-base-pair flanking positions were considered, whereas additional information from the neighboring features beyond one base pair did not result in any further improvement of the classifier (Supplementary Table 1).

A key feature of m6Anet is the ability to jointly model RNA modifications for all candidate fivemer sequences in the training data. To

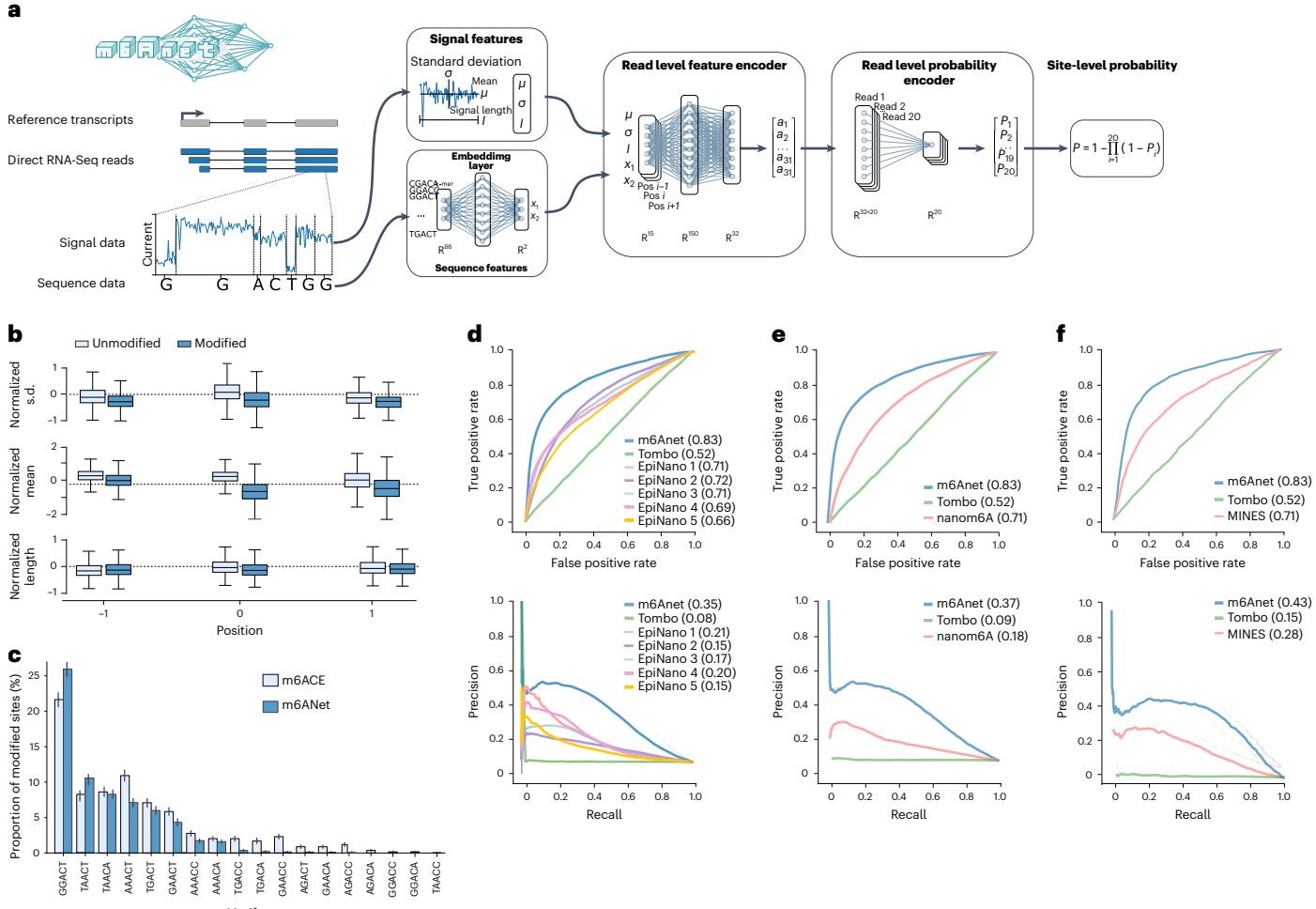

**Fig. 1 | Schematic of m6Anet and evaluation on detection of m6A in human cell lines. a**, m6Anet model schematics. **b**, Box plot showing the difference in average features distribution between different m6Anet prediction with $n = 1769$ predicted modified sites and $n = 5031$ predicted unmodified sites for the GGACT fivemer motif. The horizontal lines on the boxes show minima, 25th percentile, median, 75th percentile, and maxima. Points that do not fall within 1.5× of the interquartile range are considered outliers and are not shown on the plot. **c**, Comparison of the proportion of modified sites predicted as modified by m6Anet and by m6ACE on the DRACH fivemer motifs. The bar plot center

represents the proportion of modified sites for each fivemer motif while the error bar represents the estimated 95% confidence interval around the center values with a total of $n = 5,579$ for m6ACE modified positions, $n = 4,784$ for m6Anet-predicted modified positions and $n = 121,853$ for m6ACE unmodified positions and $n = 122,648$ for m6Anet-predicted unmodified positions. **d**, ROC curve (top) and PR curve (bottom) of m6Anet against all five EpiNano models and Tombo. **e**, ROC curve (top) and PR curve (bottom) of m6Anet against nanom6A and Tombo. **f**, ROC curve (top) and PR curve (bottom) of m6Anet against MINES and Tombo.

evaluate if this approach biases the prediction of m6A sites based on the sequence, we compared the fivemer frequency of predicted m6A sites with the fivemer frequency observed in m6ACE-Seq data on positions that have not been used to train m6Anet model parameters. We find that m6Anet predictions have a comparable fivemer profile as the m6ACE-Seq data, with less frequent motifs being equally represented (Fig. 1c), showing that m6Anet captures the expected modification rates per fivemer from a single model that combines features from signal and sequence.

### m6Anet accurately identifies m6A sites

To evaluate the performance of m6Anet we tested the model on direct RNA-Seq data from the HEK293T cell line[43], using m6ACE-Seq[28] and miCLIP data[26] from the same cell line as ground truth. We then compared the performance of m6Anet against EpiNano[45,46], MINES[47], Tombo[38], and nanom6A[48] using the area under the curve (AUC) of the receiver operating characteristic (ROC) and precision–recall (PR) curves to quantify the model accuracy. On the HEK293T cell line, m6Anet achieves a ROC

AUC of 0.83 and PR AUC of 0.35 (Fig. 1d and Supplementary Table 2). Among the other methods, only EpiNano and Tombo return predictions for all DRACH motifs, however, at a lower accuracy compared to m6Anet (Fig. 1d). Since MINES and nanom6A output predictions only for 4 and 12 fivemers respectively, we ran separate validation between MINES, nanom6A, and m6Anet on these motifs. On these data, m6Anet achieved a ROC AUC of 0.83 (4 motifs and 12 motifs) and a PR AUC of 0.43 (4 motifs) and 0.37 (12 motifs) outperforming both MINES and nanom6A (Fig. 1e,f and Supplementary Table 2), suggesting that m6Anet provides the most accurate predictions of candidate m6A among existing methods.

### m6Anet generalizes to new cell lines and species

In order to test how well m6Anet generalizes to data from a new cell line or different species, we evaluated m6Anet on two human cell lines (HEK293T and Hct116) and one *Arabidopsis* dataset[54]. For training and testing we used m6A site predictions from m6ACE-Seq and miCLIP for the human cell lines. For the *Arabidopsis* data, we obtained m6A site

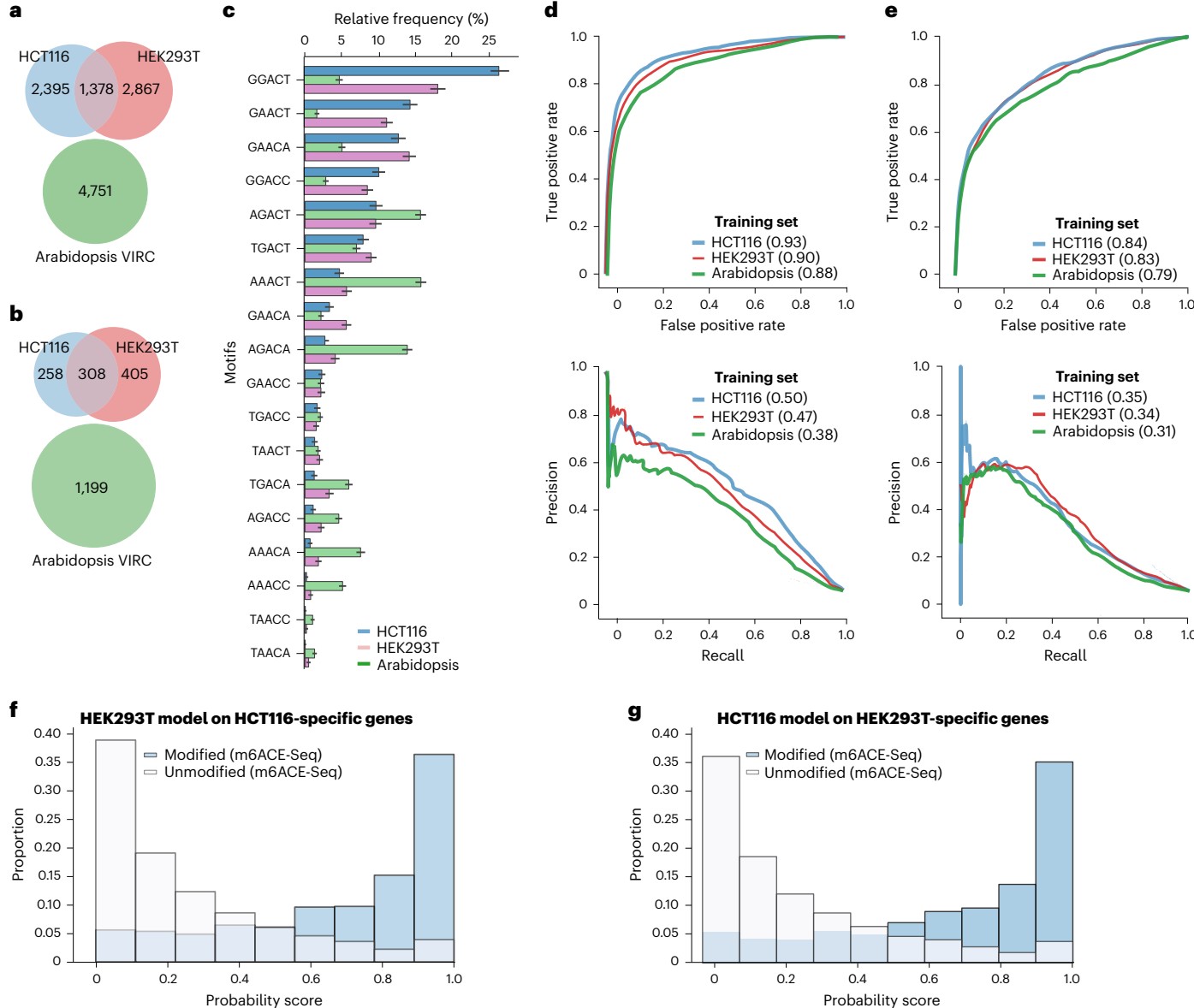

**Fig. 2 | Comparison of m6Anet model across two different cell lines. a,b,** Distribution of modified positions across all three cell lines on the training (**a**) and the test (**b**) sets (area shown to proportion). **c**, Bar plot comparing the relative proportion of methylated motifs for the HCT116, HEK293T, and Arabidopsis VIR-1 complemented mutant (VIRC). The bar plot center represents the proportion of predicted modified sites for each fivemer motif over all predicted modified fivemer sites while the error bar represents the estimated 95% confidence interval around the center values with n = 5,475 for HCT116 predicted modified positions, n = 6,804 for HEK293T predicted modified positions, and n = 13,089

for VIRC predicted modified positions. **d**, ROC curve (top) and PR curve (bottom) of the models trained on the HCT116 train set, HEK293T train set, and *Arabidopsis* training set, and tested on the HCT116 Test set. **e**, ROC curve (top) and PR curve (bottom) of the models trained on the HCT116 training set and HEK293T training set, and *Arabidopsis* training set, and tested on the HEK293 test set. **f,g,** Distribution of probability score of HEK293T (HCT116) model on the genes that are expressed only on the HCT116 cell test set (HEK293T test set). Histogram shows that m6Anet trained on both cell lines can make accurate predictions on a set of genes that are not present in their original training data.

predictions by comparing direct RNA-Seq data with low m6A modification (VIR-1 knockout (KO)) against a control (VIR-1 complement) using xPore[43], which we combined with the site-level labels from Parker et al. (Methods). For this comparison, we split the dataset on the gene level into a training and test set, ensuring that the test sets on the human cell lines comprised the same genes (Fig. 2a,b). A comparison of the relative frequency of methylated *k*-mers in the human and *Arabidopsis* data shows that both species use substantially different m6A *k*-mer profiles (Fig. 2c). Motifs such as GGACT, GAACT, and GGACA that are the most prevalently methylated in the human cell lines make up a much smaller proportion of methylated motifs in the *Arabidopsis* data.

By contrast, the most frequently methylated *k*-mers in *Arabidopsis* are AAACT, AGACT, and AGACA, which are less frequently methylated in the human m6A data, thereby providing a scenario to evaluate the generalizability of m6Anet to new data with substantially different modified *k*-mer frequencies.

When we compared the three pre-trained models, we found that they generated predictions with a similar accuracy, even when the models were trained on a different species using largely different modified *k*-mer profiles (Fig. 2d,e, Supplementary Fig. 2a–e, and Supplementary Tables 3–5). The predicted m6A sites display a strong 3′ untranslated region (UTR) enrichment that is typical for m6A and

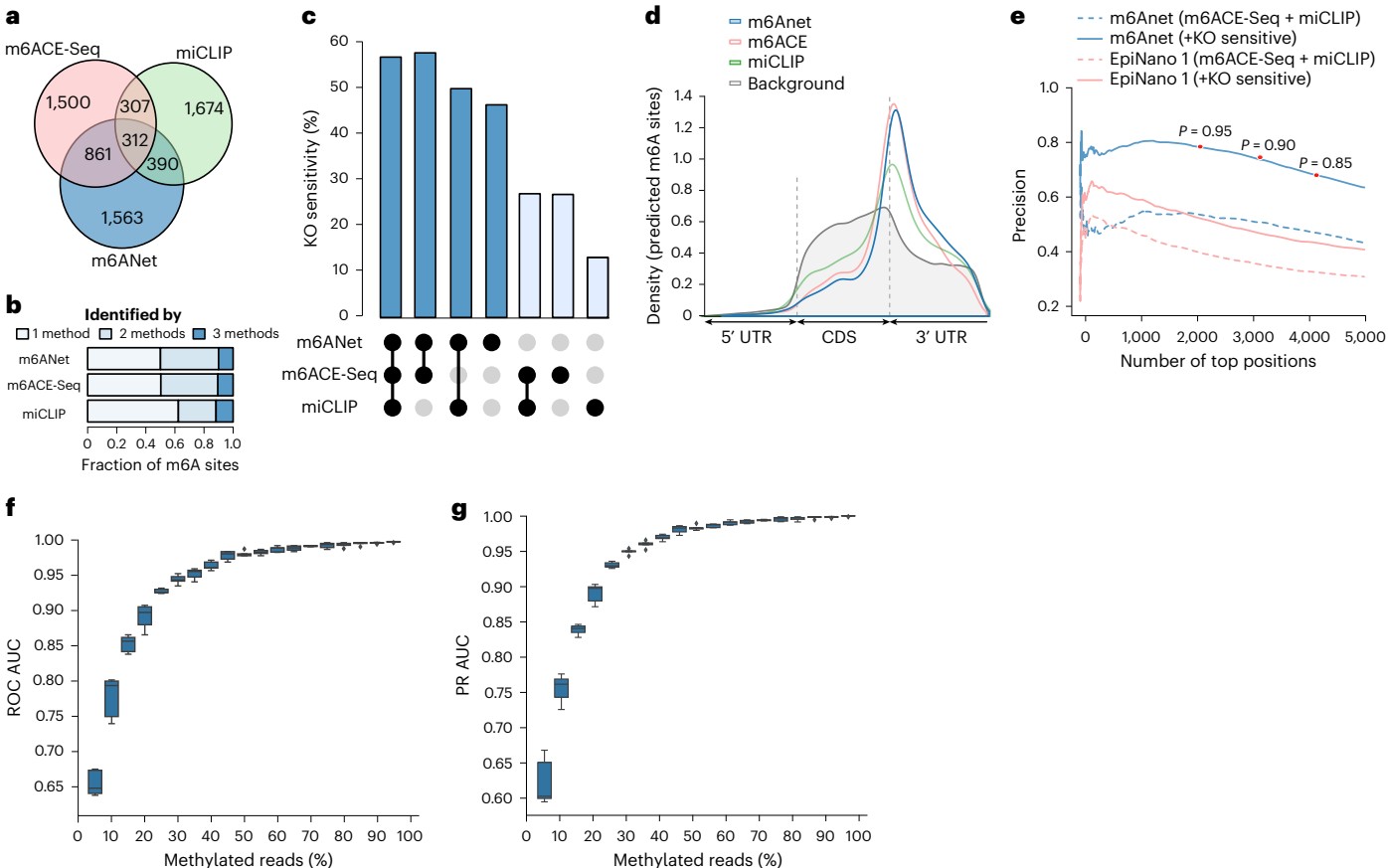

**Fig. 3 | Performance comparison between m6Anet, m6ACE-seq, and miCLIP on HEK293T cell line. a,b,** Total number of modified sites captured by m6Anet, m6ACE-seq and miCLIP (area in **a** shown to proportion). **c,** Percentage of captured sites that a show significant shift in signal distribution against METTL3-KO for each of the three protocols. Statistical significance is defined by $P < 0.05$ after multiple testing correction using Benjamini–Hochberg procedure where the $P$ value is quantified using xPore. **d,** Metagene plot of the modified sites captured by the three protocols against the background distribution of all DRACH sites in the data that have at least 20 reads. **e,** The adjusted precision after including position sensitive to METTL3-KO of m6Anet and EpiNano. Red dots indicate the precision at different probability thresholds. **f,g,** ROC AUC and PR AUC of m6Anet on different mixtures of methylated and unmethylated direct RNA-Seq reads from synthetic sequences with $n = 5$ corresponding to five different random mixings of the synthetic and IVT reads for each methylation level. The horizontal lines on the boxes show minima, 25th percentile, median, 75th percentile, and maxima. Points that do not fall within 1.5× the interquartile range are considered outliers and are not shown on the plot.

which is similar between the models trained on human and *Arabidopsis* (Supplementary Figs 2f,h). Noteworthy, the performance of m6Anet trained on *Arabidopsis* and tested on human cell lines is still better than other methods (Fig. 1d–f and Supplementary Fig. 2i–n). On the HCT116 cell line, the model learned on the HEK293T data even shows a better performance than on the original cell line used for training, indicating that the training procedure in m6Anet generates robust, generalizable models (Fig. 2d,e). Furthermore, m6Anet was able to identify m6A sites on genes that are not expressed in the cell lines used for training (Fig. 2f,g). These data demonstrate that m6Anet generalizes robustly to other cell lines without a loss in accuracy owing to cell-type-specific data. While a species-specific model will provide best results, in the absence of a species-specific training data, m6Anet still provides accurate predictions even when the default human-trained model is used.

## m6Anet-specific predictions are sensitive to METTL3 knockout

While the overall accuracy for detection of m6A from direct RNA-Seq data is high, many m6A sites predicted by m6Anet are not identified by these experimental approaches. Different methods for profiling m6A have been described to identify different sets of m6A sites[28]. Indeed, in the HEK293T cell line, the largest number of sites are detected by only

one protocol (Fig. 3a). Among the three protocols, m6Anet predictions show an equal or higher fraction of support by other technologies (Fig. 3b). To evaluate whether technology-specific m6A site predictions are valid m6A sites, we identified positions that are sensitive to loss of the m6A writer METTL3. Using an existing approach for comparative analysis of direct RNA-Seq (xPore), we mapped m6A sites in the HEK293T cell line by comparing it against a METTL3-KO cell line that is depleted of m6A[28,43]. We then define DRACH sites that have a significant difference compared to this control as KO-sensitive sites, resulting in 1,888 candidate positions identified by xPore (Supplementary Table 2; Methods). The sites that are detected by m6Anet, m6ACE-Seq, and miCLIP show the highest fraction of KO-sensitive sites (57%; Fig. 3c). Among the sites that are only detected by one method, m6Anet predictions have the highest proportion of KO sensitivity detected by *xPore* (46%; Fig. 3c), with a less stringent method to define KO-sensitive sites further increasing the fraction for all three protocols (Supplementary Fig. 3a). As the usage of a direct RNA-Seq-based method for evaluation might favor m6Anet predictions, we also investigated the enrichment of m6A positions along the transcript coordinates. This analysis shows that all the sites that are captured by the three methods are enriched in the 3′ end of the coding sequence as expected for m6A (Fig. 3d). m6A sites that are only found in one method show a

similar pattern, with m6ACE-Seq and m6Anet predictions showing the strongest enrichment (Supplementary Fig. 3b), suggesting that many of these technology-specific m6A predictions are indeed valid m6A-methylated positions.

### m6Anet achieves high precision among top predicted sites

As the different methods generate different m6A site predictions, they might underestimate the precision when used for evaluation. Indeed, including the additional METTL3-KO-sensitive m6A sites into the validation set increases the estimated precision for m6Anet and other methods based on direct RNA-Seq (Fig. 3e, Supplementary Fig. 3c, and Supplementary Table 2). Using these additional labels indicates that the *Arabidopsis* model achieves almost identical precision to the human-trained models (Supplementary Fig. 3d). At a threshold of 0.9, m6Anet achieves a precision of 70.5%, which is significantly higher than the estimate based on m6ACE-Seq or miCLIP alone (Fig. 3e). Yet, as even these data might contain incomplete modification labels, we estimated the precision of m6Anet using synthetic sequences where modification labels are complete. Here we used synthetic sequences of m6A-modified libraries and unmodified in vitro transcribed (IVT) RNA libraries that were sequenced using direct RNA-Seq[45]. We then combined modified and unmodified reads at specific ratios to simulate synthetic data for a wide spectrum of modification rates (Methods) and applied the m6Anet model trained on the human cell line to predict m6A sites on these synthetic sequences. We observe a near optimal accuracy with at least 50% modified reads (ROC AUC > 0.98; Fig. 3f,g). With modification rates between 25% and 50% m6Anet still achieves highly accurate classification (AUC > 0.93; Fig. 3f,g). While the number of false positives increases with lower modification rates, the recommended threshold of 0.9 still achieves perfect precision (100%) even with just 5% modified reads (Supplementary Fig. 3e). These results confirm that the precision of m6Anet is underestimated when comparing it to labels obtained from miCLIP or m6ACE-Seq, with many novel sites identified by m6Anet most likely reflecting valid, technology-specific m6A predictions.

### m6Anet provides single-molecule m6A predictions

While the primary output of m6Anet is a site-level modification probability, it was designed to learn a hyper-dimensional representation of each read on the basis of its signal and sequence features, which is then used to infer a read-level modification probability. To evaluate the ability of m6Anet to discriminate between modified and unmodified reads we estimated the read-level probability for the synthetic data. On these data m6Anet achieves a ROC AUC of 0.90 and a PR ROC of 0.91, suggesting that m6Anet accurately identifies single-molecule m6A modifications (Fig. 4a,b).

To illustrate the ability to predict per-molecule modifications in human cell line data, we extracted the read-level representation and probabilities from both the HEK293T wild-type and KO cell lines for candidate m6A positions ($P > 0.9$ in wild type, $P < 0.2$ in KO; Methods). We then performed a principal component analysis (PCA) on the high-dimensional read-level features to map reads into a two-dimensional space. We find that reads form two clusters that are dominated by the KO reads and wild-type reads, respectively (Fig. 4c and Supplementary Fig. 4a–h). As a control, we mapped the synthetic reads into this read-level feature map, confirming that the wild-type reads resemble m6A-modified molecules, whereas the KO reads resemble unmodified molecules (Fig. 4c). Using these clusters we projected data from individual reads for the positions identified to have the highest modification probability into this read-level feature map (Fig. 4d–f, Supplementary Fig. 4i–k, and Supplementary Table 6). While reads from the KO sample have low predicted m6A probabilities and fall into the unmodified cluster, reads from the wild-type samples are enriched in the modified cluster, demonstrating how m6Anet enables the analysis and visualization of single-molecule m6A predictions.

### Single-molecule predictions capture the m6A stoichiometry

The ability to infer single-read modification probabilities suggests that m6Anet can predict the underlying, site-level modification stoichiometry. Using the synthetic data, we selected a threshold on the read-level probability that provides the greatest difference between true positive and false positive rate for single-molecule modification predictions (Methods). We then compute the modification rate for each position as the number of modified reads per site with a read-level modification probability above that threshold. On the synthetic sequence mixtures, the estimated relative modification rate closely matches the expected modification rate (Fig. 4g).

To validate whether m6Anet captures the proportion of modified reads in human data, we analyzed direct RNA-Seq data from METTL3-KO and wild-type samples that were mixed at specific proportions corresponding to an expected relative m6A stoichiometry of 0%, 25%, 50%, 75%, and 100%[43]. We then estimated the stoichiometry on the sites that were predicted to be modified in the 100% wild-type samples ($P \ge 0.9$) and that are predicted to be unmodified in the KO samples ($P \le 0.2$) (Supplementary Fig. 5a and Supplementary Table 7). While we observe more variation compared to the synthetic sequences, the median relative modification rate closely matches the expected (Fig. 4h). Additionally, we also found that the median modification rate remains close to the expected modification level on sites with less stringent threshold ($P \ge 0.7$ in 100% wild type, $P \le 0.4$ in 100% KO) and across transcripts with different RNA localizations[55] (Supplementary Fig. 5b–h), suggesting the robustness of our stoichiometry estimates. This is also reflected in the single-molecule predictions, where we observed a gradual shift of reads from the modified cluster to the unmodified cluster, corresponding to the expected changes in the relative m6A stoichiometry (Fig. 4i–k and Supplementary Fig. 5i,j). These data suggest that m6Anet captures variation in the underlying modification rates that can be used to compare sites within one sample, or to estimate site-specific and global differences in m6A abundance across multiple samples or conditions.

### Discussion

Supervised approaches promise to enable the accurate detection of RNA modifications from direct RNA-Seq data. These methods rely on accurate training data, which can be obtained through experimental protocols to identify RNA modifications such as m6ACE-Seq or miCLIP, using synthesized RNAs that contain specific modifications of interest, or from a comparative analysis of direct RNA-Seq data. However, these methods only provide site-level modification labels, whereas Nanopore data is provided for individual RNA molecules for which the modification status is not observed. Here we address this by developing m6Anet, a neural-network-based MIL model. m6Anet combines learning the representation of individual reads with classifying m6A-modification sites, outperforming existing computational methods and achieving an accuracy that is comparable to experimental approaches.

One of the key challenges is the quantification of transcriptome-wide modification rates. The ability to quantify modification stoichiometry from direct RNA-Seq data has been demonstrated by comparative approaches such as xPore[56] for m6A modification and nanoRMS[49] for pseudouridine. m6Anet, on the other hand, outputs the single-molecule modification probability from a single sample. Owing to the MIL framework, this is achieved without single-molecule modification labels, enabling the use of much larger datasets compared to other single-sample methods that require synthetic data[48]. With single-molecule predictions, m6Anet enables not only the quantification of the site-level modification stoichiometry without a control sample, but also facilitates insights into the relation of read and transcript-level features such as polyadenylation, degradation, or alternative splicing with RNA modifications.

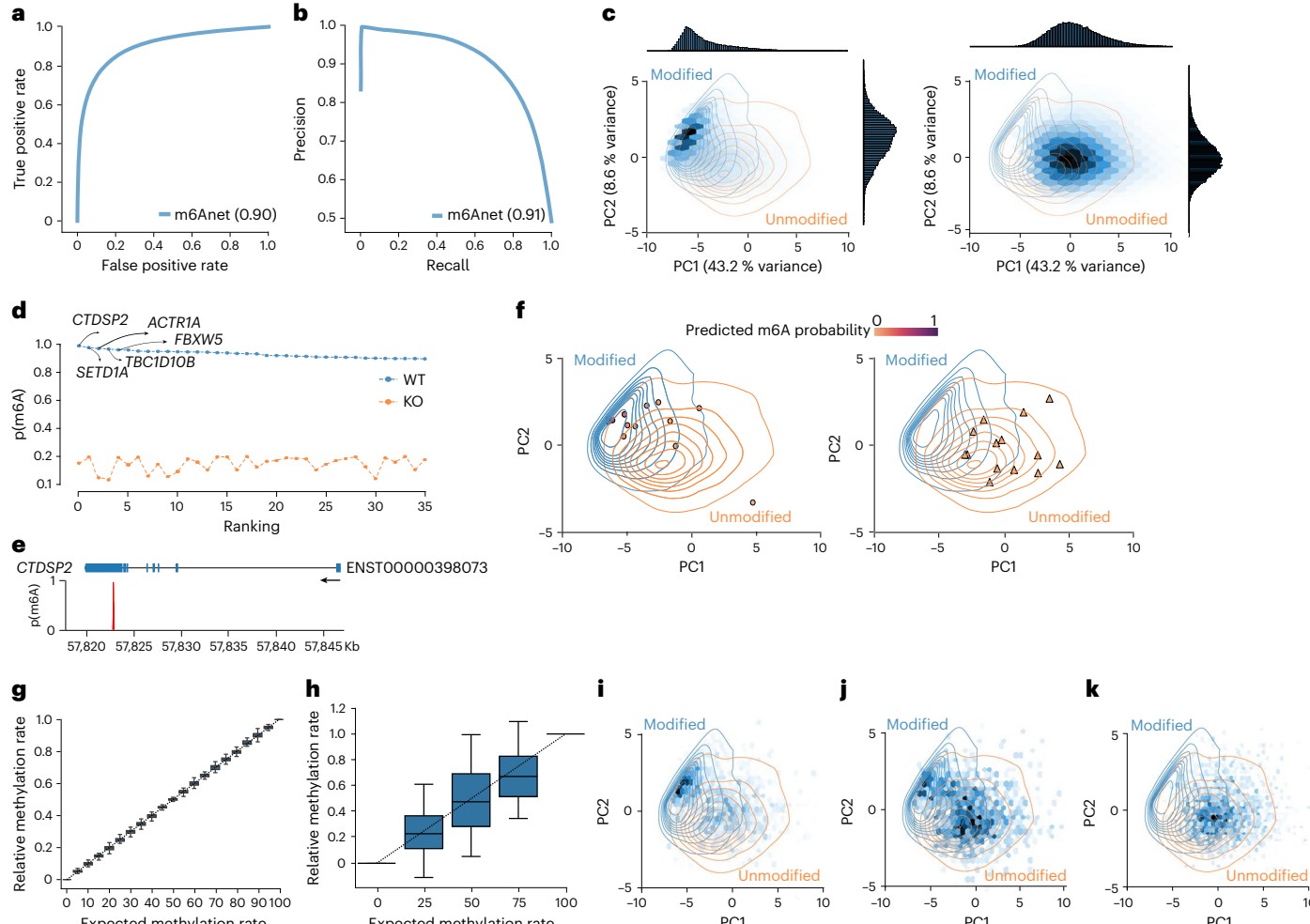

**Fig. 4 | Quantification of m6Anet on HEK293T cell line. a,b,** ROC curve (**a**) and PR curve (**b**) of m6Anet single-molecule prediction on IVT and synthesized RNA reads from curlcake datasets. **c,** Hex plot of the top four modified fivemer (GGACT, GAACT, GGACA, AGACT) for the wild-type sample (left) and KO sample (right) with density plot of modified and unmodified samples from curlcakes reads. **d,** Ranking plot of the positions in **c** and the genes associated with the top positions. **e,** Gene plot of the top ranked modified position. **f,** Scatter plot of randomly sampled reads from the position highlighted in **e**, on the wild-type cell line (left) and METTL3-KO cell line (right). The colour indicates the read-level m6A probability. **g,h,** Box plots showing the predicted relative methylation rate on synthetic reads with $n = 43$ (**g**) and on the HEK293T cell line with different levels of wild-type and METTL3-KO RNA mixtures with $n = 34$ (**h**) (Methods). The horizontal lines on the boxes show minima, 25th percentile, median, 75th percentile, and maxima. Points that do not fall within 1.5× the interquartile range are considered outliers and are not shown on the plot. **i–k,** Hex plots of the read-level feature map for 0% (**i**), 50% (**j**), and 100% (**k**) KO mixtures on filtered positions.

Even though m6Anet was designed to handle missing read-level modification information, it still relies on the accuracy of site-level training data. Depending on how these data were generated, such labels could be incomplete[57,58], or include multiple distinct modifications[26,28] thereby introducing noise in the training data and a reduction in the model performance. Here we find that the prediction accuracy on m6A appears to be high even when different training datasets are used. Nevertheless, additional training data on different modifications and experimental protocols will likely further improve the prediction accuracy for supervised approaches such as m6Anet.

While supervised methods can identify RNA modifications in a single sample, comparative methods facilitate the analysis across conditions[41,43,59]. One of the key advantages of supervised methods over comparative methods is their ability to predict the occurrence of specific RNA modifications such as m6A. By predicting m6A modifications on candidate sites identified by comparative methods, m6Anet can overcome their inability to assign specific modification types, thereby facilitating modification-specific analysis of differential modifications.

In contrast to short-read-based experimental approaches for profiling RNA modifications, direct RNA-Seq is a simple assay that can make m6A profiling scalable. However, similar to experimental protocols that are influenced by aspects such as antibody-specificity[26,28], the accuracy of m6Anet will be influenced by aspects such as the sequencing chemistry, base-calling algorithms or accuracy in the alignment of reference sequence to signal. Additionally, improvement in the pore chemistry might require m6Anet to be retrained in order to take advantage of such changes. Further improvements in the sequencing technology and methods that extract summarized data from Nanopore signals can further increase the accuracy of m6Anet. While we observe a high number of technology-specific m6A predictions, our data supports that these are likely valid m6A sites.

Here we applied m6Anet to identify m6A modifications; however, it was designed to facilitate training on other RNA modifications of interest as well. While m6Anet could be used to identify other naturally occurring RNA modifications, it can also be trained to predict artificial modifications that help to identify single-molecule RNA structures[60]

after retraining. Moreover, it will also complement existing experimental approaches by increasing confidence and resolution, enabling the accurate prediction of site-level modification while facilitating the additional exploration of single-molecule modification probabilities from a single run of direct RNA-Seq data.

## Online content

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

## Methods

m6Anet performs detection of RNA modification using direct RNA-Seq data by formulating it as a MIL problem. Each position corresponds to a $k$-mer sequence $S$ of length $k = 5$ with:

$$S_i = \{s_{i-2}, s_{i-1}, s_i, s_{i+1}, s_{i+2}\} \tag{1}$$

Here $S_i \in \{A, C, G, U\}$ corresponds to the nucleotide of position $i$. For each position, the site modification status is given by $y_i$ where

$$y_i = \{1 \text{ if position i is modified, } 0 \text{ otherwise}\} \tag{2}$$

We also assume that each read $j$ at position $i$ has a modification status described by $y_{i,j}$ given by:

$$y_{i,j} = \{1 \text{ if read j at position i is modified, } 0 \text{ otherwise}\} \tag{3}$$

While $y_i$ can be observed, $y_{i,j}$ cannot be observed and remains unknown. Each read $j$ at position $i$ is described by the feature vector $x_{ij} \in R^{15}$ with:

$$x_{i,j} = \{\mu_{i-1,j}, \mu_{i,j}, \mu_{i+1,j}, \sigma_{i-1,j}, \sigma_{i,j}, \sigma_{i,j+1}, l_{i-1,j}, l_{i,j}, l_{i+1,j}, \\ f(S_{i-1})_1, f(S_{i-1})_2, f(S_i)_1, f(S_i)_2, f(S_{i+1})_1, f(S_{i+1})_2\} \tag{4}$$

where $\mu_{i,j}$ represents the normalized mean nanopore raw signal of read $j$ at position $i \sigma_{i,j}$ represents the normalized standard deviation of the nanopore raw signal of read $j$ at position $i$, and $l_{i,j}$ represents the normalized dwelling time of read $j$ at position $i$. Furthermore, we encode all $N_S$ possible fivemer sequence motifs $S$ that are included in the training data into a two-dimensional vector using a neural network embedding layer $f : N_S \rightarrow R^2$, with $N_S = 66$ in the case of m6A (DRACH). Thus, the quantity $f(S_i)_k$ gives the $k$-th dimension of the embedded vector of the fivemer motif $S_i$, with $k \in \{1, 2\}$. Each position $i$ with $N_i$ reads is then described by

$$X_i = \{x_{i,1}, x_{i,2}, \ldots, x_{i,N_i}\} \tag{5}$$

In the first step, m6Anet estimates the read-level modification probability $p_{i,j}$ of the read $j$ at position $i$ being modified:

$$p_{i,j} = PrPr(x_{i,j}) = F(x_{i,j}) \tag{6}$$

Where $F : R^{15} \rightarrow R$ is parameterized by a neural network with two hidden layers of dimension 150 and 32, respectively. In the second step, m6Anet pools the read-level probability using a noisy-OR pooling layer to estimate the site-level modification probability $p_i$:

$$P_i = PrPr(p_{i,1}, p_{i,2}, \ldots, p_{i,N_i}) = 1 - \prod_{j=1}^{N_i}(1 - p_{i,j}) \tag{7}$$

The noisy-OR pooling layer captures the assumption that a site is modified if at least one of its reads is modified. In practice, the noisy-OR pooling layer encourages any gradient-based learning methods to update the model parameters with respect to all reads instead of just a single modified read. As a result, the site probability estimated by m6Anet should reflect the changes in the number of modified reads between different sites.

To train the network, we minimize the average cross entropy loss $\mathcal{L}$ between $p_i$ and $y_i$ for all sites

$$L = \frac{1}{N} \sum_{i=1}^{N} y_i \log\log P_i + (1 - y_i)(1 - \log\log P_i) \tag{8}$$

Here $f$ and $F$ are learnt in an end-to-end fashion by minimizing the cross entropy loss $L$ with the Adam optimizer. Consequently, the network learns to predict the individual read probability $p_{i,j}$ along with

optimized sequence representation $f(N^S)$ that will minimize the discrepancy between $p_i$ and $y_i$ with respect to the noisy-OR pooling layer. We have evaluated alternative pooling layers, such as the Attention and gated Attention-based pooling[61] but have not found any statistically significant improvement in the performance of m6Anet compared to the noisy-OR pooling layer for m6A detection.

### Preprocessing for m6Anet

m6Anet requires the output from Nanopolish eventalign function[52] order to group continuous Nanopore current measurements from each read into events and map them to their corresponding positions in the transcriptome. Each nanopolish event comprises the mean, standard deviation, and dwelling time of its constituting raw signals and since multiple events can be assigned to the same location in the transcriptome, m6Anet then takes a weighted average of each of these features on the basis of the size of their respective groups. Afterwards, m6Anet discards positions with mismatched fivemers and computes the mean and standard deviation of the signal features for each possible fivemer motif across the transcriptome. Lastly, m6Anet performs $z$-normalization on the weighted average features on the basis the mean and standard deviation of the fivemer motif of the given segment. The preprocessing function is implemented in m6Anet using functions from pandas 1.2.5 and numpy 1.20.3.

### Data processing

**Processing of direct RNA-Seq data.** All data used in this work was obtained from refs. [43,45,53,54]. To train and validate m6Anet, we downloaded a single replicate (replicate 2 run 1) of the HCT116 cell line and a single replicate of the HEK293T cell line (replicate 1) while to run xPore, we downloaded all replicates of the HEK293T cell lines as recommended. We also downloaded all four replicates of the VIR-complemented (VIRc) and VIR-1 mutants (vir-1) to run xPore and also to train and validate m6Anet. Data was base-called from the raw fast5 files using Guppy and aligned to the transcriptome with mini-map2.1 (minimap2 '-ax map-ont -uf −secondary=no') using the GRCh38 Ensembl annotations release version 91 for HCT116 and HEK293T cell lines. We used a combined FASTA file containing coding and non-coding RNA reference annotations, keeping only the transcripts that matched the reference genome annotations (nf-core/nanoseq: https://doi.org/10.5281/zenodo.3697960). *Arabidopsis* VIRc and vir-1 mutants were both aligned to the TAIR10 transcriptome (minimap2 '-ax splice -uf -k14') while curlcake datasets were aligned to the reference sequence provided by ref. [45] (minimap2 '-ax map-ont'). Afterwards, we ran Nano-polish 0.11.3 with the '--scale-events' and '--signal-index' options.

**m6A-crosslinking-exonuclease sequencing.** Modified positions for m6ACE-seq are obtained from refs. [28,43], and we also follow their preprocessing steps for the HEK293T cell lines and include only those positions that are METTL3-dependent (wild type/KO relative methylation level ratio ≥4.0, $P$ value of one-tailed $t$-test, <0.05). As for the HCT116 cell line, we consider any sites that appear in the m6ACE-seq library to be modified since the absence of METTL3-KO data means we are not able to filter on the basis of the wild type/KO relative methylation level like in the HEK293T cell lines.

**m6A individual-nucleotide-resolution crosslinking and immuno-precipitation.** Modified positions from miCLIP were obtained from ref. [26], and we combine both CIMS and CITS miCLIP libraries from the supplementary and consider a position to be modified if it is found in any of these libraries.

### Model evaluation

**Contribution of flanking regions to m6Anet performance.** To evaluate the performance of m6Anet under different combinations of features, we performed a fivefold cross validation on the HCT116 dataset.

In each fold, we train our model on 75% of our training data for 60 epochs and choose the model that performs the best on the remaining 25% of the training data and validate the performance of the model on the test set. We also ensure that no genes are shared between the training, validation, and test set during the evaluation. During training the parameters of the model are learnt by minimizing the cross entropy loss using the Adam optimizer[62] with amsgrad[63] turned on. On each site, we sample 20 reads and during test time, we run the model five times and average the probability value across the five runs. Results are shown in Supplementary Data 1. All models are implemented on Pytorch v1.7.1 (ref. [64]). Training is done with a fixed learning rate of 0.0004 and a mini-batch size of 512 on a single NVIDIA GeForce GTX 1080 Ti.

**Comparison between m6ANet and other models on HEK293T cell line.** To have a fair comparison between m6Anet and existing methods to detect m6A modifications, we performed the comparison against other models on the HEK293T cell line on a set of genes, which were not used to train the m6Anet model. We consider a position to be modified if it is captured by either miCLIP or m6ACE-Seq as modified and we only consider DRACH sites that have at least 20 reads.

**Tombo.** We ran Tombo v.1.5.1 from https://github.com/nanoporetech/tombo. To detect modifications, we first resquiggled the raw reads with tombo-resquiggle and performed de novo detection with tombo detect_modifications de_novo. Since tombo outputs a fraction of modified reads per position, we treat this as the probability of a site being modified for our comparison.

**EpiNano.** We ran EpiNano 1.1 and 1.2 from https://github.com/enovoa/EpiNano and in both cases, we excluded feature generations for positions that do not contain AC center nucleotides (without this step, the results were not returned within 7 days on a AMD EPYC 7R32 server with 180 GB of memory). There are four SVM models on EpiNano 1.1 and on SVM model on EpiNano 1.2 that could work with a single sample of direct RNA-Seq data. We numbered these models from one to five, respectively.

**MINES.** We ran MINES from https://github.com/YeoLab/MINES on cDNA mode, following the steps that are specified in the readme file on the Github page. The original MINES model does not output the probability of a site being modified but instead only shows sites that are considered modified. For this comparison, we modified the code so that the RandomForest model outputs the probability of a site being modified and we compared the results with m6Anet on sites shared between the two methods. The modified code is available at https://github.com/chrishendra93/MINES.git.

**nanom6A.** We ran nanom6A from https://github.com/gaoyubang/nanom6A. Similar to Tombo, it only outputs a fraction of reads that are modified for each site and so we treat these numbers as the probability of a site being modified.

**Comparison between m6Anet, m6ACE-Seq, and miCLIP.** To evaluate the relative performance between m6Anet and other commonly used experimental protocols, we performed a comparison with miCLIP and m6ACE on the HEK293T cell line. We set a $P = 0.9$ threshold for m6Anet site probability to select modified sites. miCLIP and m6ACE-Seq data was obtained and processed as described above. Visualization is done through the matplotlib 3.3.4 library and upsetplot 0.6.0.

To calculate whether a site is KO sensitive or not, we ran xPore 1.0 on replicate 1, 2, and 3 of the HEK293T samples provided by[43] with pooling option and a minimum read threshold of 20. To be conservative about our estimates, we imputed any sites that are not present in the xPore run with $P$ value of 1 (not differentially modified). We performed multiple test corrections using Benjamini–Hochberg procedure implemented in statsmodels 0.12.2 and set an alpha rate of 0.05.

To obtain a second (less stringent and less accurate) estimate for KO-sensitive sites we also ran Welch's $t$-test from the scipy package function ttest_ind (setting equal variance to false). Similar to the analysis with xPore, we pooled reads from all three replicates and required tested positions to have a minimum of 20 reads. We then performed multiple test corrections using Benjamini–Hochberg procedure from statsmodels 0.12.2, set an alpha rate of 0.05 and imputed any other sites that do not meet the filter criteria with a $P$ value of 1.

**Metagene plot.** To visualize the distribution of m6A sited across the transcript (metagene plot), we first mapped each gene coordinate to transcript coordinate on the basis of the most expressed transcripts per gene. Afterwards, we annotate each position on the basis of its location along the transcript as 3′ UTR, 5′ UTR, or coding sequence. We then calculate the relative position of each position on the transcript and plot the abundance of those positions that are considered modified by m6Anet, m6ACE-seq, or miCLIP.

**Comparison of m6Anet performance on HEK293T and HCT116 cell lines.** To measure the robustness of m6Anet across different cell lines, we train two different models on the HEK293T and HCT116 cell lines, respectively, and measure the performance of each model on both HEK293T and HCT116 test sets. We randomly select 500 genes that are present in both cell lines to form two test sets for both cell lines and use the remaining genes as training data. We further split 20% of the training set for each cell line at the gene level into a validation set for model selection.

**Comparison of m6Anet performance on *Arabidopsis* VIR-1 mutant.** To measure the robustness of m6Anet across species, we train an additional m6Anet model on four replicates of the VIRc mutants and compare the performance of this model against models that are trained on HEK293T and HCT116 cell lines. We randomly select 20% of all the genes expressed in the VIRc mutant to form a test set and use the remaining genes as training data. We further set aside 20% of the training genes to form a validation set for model selection.

To determine whether a site is modified or not, we included the labels provided by Parker et al.[54] and statistically significant sites obtained from running xPore diffmod on all four replicates of the VIRc and vir-1 mutants with pooling option and a minimum read threshold of 20. Additionally, we also performed multiple test corrections using Benjamini–Hochberg procedure and set an alpha rate of 0.05 for both methods and only consider a site to be modified according to either one of the methods if it passes this threshold.

**Inference on the synthetic Curlcakes dataset.** To measure the true performance of m6Anet, we perform inference on the synthetic RNA datasets provided by ref. [45]. We combine all sites from two replicates of the IVT sequences and consider them as unmodified while all sites from the two replicates of the curlcake sequences are considered to be m6A modified. Following the authors, we exclude all fivemer motifs that contain more than two adenosines and only consider DRACH motifs when measuring m6Anet performance.

To form a mixture of methylated–unmethylated reads from each site, we perform random sampling of reads from each site shared between the IVT sequences and the modified sequences. To produce a robust estimate for m6Anet performance on the mixture sequences, we run the model five times on each mixture level and each time, we perform random sampling of all the reads to form a different mixture datasets.

**Visualization of single-molecule modification probabilities**
**Principal component analysis and read-level feature map.** To learn the read-level feature map that visualizes single-molecule m6A probability predictions, we project the high-dimensional read representations of m6Anet using a PCA and visualize the first two principal

components. We sampled 100 reads from each position and extracted the 64-dimensional features generated by the second-last layer of m6Anet from each of these reads. We ran PCA from the Python package scikit-learn 1.0.2 (ref. [65]) with n_components set to 0.99 and svd_solver set to full so that the algorithm will choose the number of components that will result in total variance explained to be as close as possible to 1.

To better visualize the features that are representative of both modified and unmodified reads, we first filtered for positions that are highly modified in the WT sample ($P \geq 0.9$) or unmodified in the KO sample ($P \leq 0.2$) and which contain the fivemer motifs GGACT, GAACT, GGACA, or AGACT. These motifs are chosen because they represent the most modified fivemer motifs in the HEK293T cell lines on the basis of miCLIP annotations or m6ACE-seq annotations. We further sampled 20 reads from each of these positions in order to minimize running time. We then calculated the density plot and hex plot on both the wild-type reads and KO reads as well as the modified and unmodified reads from the curlcakes dataset on the first two principal components of the read features using Python seaborn 0.11.1 package. We then use the resulting density plot as a read-level feature map to visualize individual molecule modification probabilities.

### Quantification of m6Anet on HEK293T mixtures
**Estimation of m6A-modification stoichiometry.** To estimate the modification stoichiometry for each potential m6A site, we extract the individual read probability from all reads expressed in each candidate site and compute the average number of modified reads per site. A read is considered to be modified if its read probability exceeds a threshold of $P = 0.0333$, which was obtained by maximizing the difference between the true positive rate and the false positive rate on the read level from the ROC curve generated from the curlcakes dataset.

**Analysis of wild type–METTL3-KO mixture samples.** To analyze the ability of m6Anet to estimate m6A stoichiometry we used the wild type–METTL3-KO mixtures from three[43] that have an expected relative average modification rate of 0% (METTL3-KO), 25%, 50%, 75%, and 100% (wild type). We filter for those positions that are present in all samples and are either fully modified (probability greater than 0.9 in the 100% wild-type sample) or not modified (probability less than 0.2 in the KO samples) Afterwards, we normalize the predicted modification rate for each site to obtain the relative modification rate with respect to the wild-type cell line (100%). Since we expect different levels of baseline methylation from each site in the METTL3 wild-type dataset, we subtract the predicted modification rate of each site by the corresponding predicted modification rate in the METTL3-KO site, and normalized this by the observed difference between the KO and wild-type sample. Specifically, the relative methylation rate for site $i$ is then calculated as:

$$\text{Relative Methylation Rate}_i = \frac{\text{Methylation Rate}_i - \text{KO Methylation Rate}_i}{\text{WT Methylation Rate}_i - \text{KO Methylation Rate}_i}$$

**Analysis of stoichiometric performance for different RNA biotype.** We also investigate the effectiveness of our stoichiometric estimate on different RNA localization. Here we select a lower threshold for both modified and unmodified sites (probability greater than 0.7 in the 100% wild-type sample and probability lower than 0.4 in the 100% KO sample) to recover more sites for comparison and derive each gene localization using RNALocate database 13.

### Reporting summary
Further information on research design is available in the Nature Research Reporting Summary linked to this article.

### Data availability
The HCT116 cell lines data were obtained from the Singapore Nanopore Expression Project[53] through https://github.com/GoekeLab/ sg-nex-data (ENA PRJEB44348) while the HEK293T cell lines data along with its KO variants and KO mixture variants were obtained from through ENA (PRJEB40872). The *Arabidopsis* Virilizer-1 complemented mutant is obtained from the work of Parker et al.[54] and is available through ENA (PRJEB32782) while the curlcakes dataset is from Liu et al.[45] and is available at the GEO database (GSE124309).

### Code availability
The source code for m6Anet is available at https://github.com/ GoekeLab/m6anet. Installation instructions and online documentation is available at https://m6anet.readthedocs.io/en/latest/. The code to reproduce results in this manuscript is available through Code Ocean at https://codeocean.com/capsule/4723237/tree.

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

### Acknowledgements
We would like to thank A. Sim (Genome Institute of Singapore) for his helpful suggestions on the RNA biotype analysis. C.H. is supported by funding from the Institute of Data Science, National University of Singapore and NUS Graduate School—Integrative Sciences and Engineering Programme. J.G. is supported by funding from the Agency for Science, Technology and Research (A*STAR), Singapore, and by the Singapore Ministry of Health's National Medical Research Council under its Individual Research Grant funding scheme.

### Author contributions
C.H., A.T., and J.G. designed the research, developed the methods, and performed all data analysis. C.H. wrote the software with contributions from Y.K.W. C.H. and J.G. wrote the documentation. C.H. and P.N.P. preprocessed the data. P.N.P. and W.S.S.G. contributed to the interpretation of the data. C.H. and J.G. wrote the manuscript with input from A.T. and W.S.S.G. All authors approved the final manuscript.

### Competing interests
J.G. received reimbursement for travel and accommodation from Oxford Nanopore Technologies to present at the Nanopore Community Meeting in San Francisco in 2018. W.S.S.G. is an inventor on a filed provisional patent application in Singapore on the use of photo-crosslinking RNA-modification-specific antibodies and exoribonucleases to sequence RNA modifications, and owns shares in Oxford Nanopore Technologies. The remaining authors declare no competing interests.

### Additional information

# Article

**Correspondence and requests for materials** should be addressed to Alexandre Thiery or Jonathan Göke.

# Reporting Summary

Nature Research wishes to improve the reproducibility of the work that we publish. This form provides structure for consistency and transparency in reporting. For further information on Nature Research policies, see our Editorial Policies and the Editorial Policy Checklist.

## Statistics

For all statistical analyses, confirm that the following items are present in the figure legend, table legend, main text, or Methods section.

| n/a | Confirmed | |
|---|---|---|
| ☐ | ☒ | The exact sample size (*n*) for each experimental group/condition, given as a discrete number and unit of measurement |
| ☐ | ☒ | A statement on whether measurements were taken from distinct samples or whether the same sample was measured repeatedly |
| ☐ | ☒ | The statistical test(s) used AND whether they are one- or two-sided *Only common tests should be described solely by name; describe more complex techniques in the Methods section.* |
| ☒ | ☐ | A description of all covariates tested |
| ☐ | ☒ | A description of any assumptions or corrections, such as tests of normality and adjustment for multiple comparisons |
| ☐ | ☒ | A full description of the statistical parameters including central tendency (e.g. means) or other basic estimates (e.g. regression coefficient) AND variation (e.g. standard deviation) or associated estimates of uncertainty (e.g. confidence intervals) |
| ☐ | ☒ | For null hypothesis testing, the test statistic (e.g. *F*, *t*, *r*) with confidence intervals, effect sizes, degrees of freedom and *P* value noted *Give P values as exact values whenever suitable.* |
| ☒ | ☐ | For Bayesian analysis, information on the choice of priors and Markov chain Monte Carlo settings |
| ☒ | ☐ | For hierarchical and complex designs, identification of the appropriate level for tests and full reporting of outcomes |
| ☒ | ☐ | Estimates of effect sizes (e.g. Cohen's *d*, Pearson's *r*), indicating how they were calculated |

*Our web collection on statistics for biologists contains articles on many of the points above.*

## Software and code

Policy information about availability of computer code

| Data collection | No software is used for data collection |
|---|---|
| Data analysis | pandas 1.2.5, numpy 1.20.3, matplotlib 3.3.4, seaborn 0.11.1, pytorch 1.8.1 +cu102, statsmodels 0.12.2, Tombo 1.5.1, EpiNano 1.1, EpiNano 1.2, MINES (https://github.com/YeoLab/MINES), MINES modified (https://github.com/chrishendra93/MINES.git), nanom6A (https://github.com/gaoyubang/nanom6A) , m6anet 0.0.2,m6anet 1.1.0, tensorflow 2.8.0, upsetplot 0.6.0, scikit-learn 1.0.2 |

For manuscripts utilizing custom algorithms or software that are central to the research but not yet described in published literature, software must be made available to editors and reviewers. We strongly encourage code deposition in a community repository (e.g. GitHub). See the Nature Research guidelines for submitting code & software for further information.

## Data

Policy information about availability of data

All manuscripts must include a data availability statement. This statement should provide the following information, where applicable:
- Accession codes, unique identifiers, or web links for publicly available datasets
- A list of figures that have associated raw data
- A description of any restrictions on data availability

The HCT116 cell lines data were obtained through ENA (PRJEB44348) while the HEK293T cell lines data along with its KO variants and KO mixture variants were through ENA (PRJEB40872). The Arabidopsis Virilizer-1 complemented mutant is available through ENA (PRJEB32782) while the curlcakes dataset is available at the GEO database (GSE124309).

# Field-specific reporting

Please select the one below that is the best fit for your research. If you are not sure, read the appropriate sections before making your selection.

☒ Life sciences ☐ Behavioural & social sciences ☐ Ecological, evolutionary & environmental sciences

For a reference copy of the document with all sections, see nature.com/documents/nr-reporting-summary-flat.pdf

# Life sciences study design

All studies must disclose on these points even when the disclosure is negative.

| | |
|---|---|
| Sample size | We trained and test our model on a total of 10 replicates from HCT116 cell line, HEK293T cell line(1 replicate), HEK293T(1 replicate) with METTL3-KO mixture cell lines (4 cell lines in total) and Arabidopsis VIRC mutants (4 replicates). No sample size calculation was performed, all samples used here have been published before and are publicly available. Each sample contains roughly a million reads,, covering a total of two species which we feel is sufficient to demonstrate the generalisability of our approach |
| Data exclusions | Data exclusions involve filtering for positions that have less than 20 reads being expressed. This is to reduce prediction noise that can arise from making predictions on sites very few reads |
| Replication | We have set fixed random seed during training and testing of the data. Code is also available online to reproduce our experiments |
| Randomization | For each cell line involved in the study, we randomly form training, validation, and testing set, ensuring that each set contains different genes so as to ensure generalizability of the study. Model is then trained on the training set while validation set is used for model selection and test set is used to compare the performance of the models used in this study. |
| Blinding | Blinding is not relevant to this study as training set and test sets are randomly sampled and we also perform validation on all samples involved by comparing the performance of each model with comparable model trained on the test cell line, thereby ensuring impartiality in our experiments |

# Reporting for specific materials, systems and methods

We require information from authors about some types of materials, experimental systems and methods used in many studies. Here, indicate whether each material, system or method listed is relevant to your study. If you are not sure if a list item applies to your research, read the appropriate section before selecting a response.

### Materials & experimental systems

| n/a | Involved in the study |
|---|---|
| ☒ | ☐ Antibodies |
| ☒ | ☐ Eukaryotic cell lines |
| ☒ | ☐ Palaeontology and archaeology |
| ☒ | ☐ Animals and other organisms |
| ☒ | ☐ Human research participants |
| ☒ | ☐ Clinical data |
| ☒ | ☐ Dual use research of concern |

### Methods

| n/a | Involved in the study |
|---|---|
| ☒ | ☐ ChIP-seq |
| ☒ | ☐ Flow cytometry |
| ☒ | ☐ MRI-based neuroimaging |

