## [Peer Review File · Nature Methods]

Peer Review Information

Manuscript Title: Detection of m6A from direct RNA sequencing using a Multiple Instance Learning framework

Corresponding author names: Alexandre Thiery, Jonathan Göke

Editorial Notes:

Redactions – unpublished data	Parts of this Peer Review File have been redacted as indicated to maintain the confidentiality of unpublished data.
Redactions – confidential patient information	Parts of this Peer Review File have been redacted as indicated to maintain patient confidentiality.
Redactions – published data	Parts of this Peer Review File have been redacted as indicated to remove third-party material.

Reviewer Comments & Decisions:

Decision Letter, initial version:

Subject: Decision on Nature Methods submission NMETH-A47051A
Message: 7th Dec 2021

Dear Jonathan,

Your Article, "Detection of m6A from direct RNA sequencing using a Multiple Instance Learning framework", has now been seen by 4 reviewers. As you will see from their comments below, although the reviewers find your work of considerable potential interest, they have raised a number of concerns. We are interested in the possibility of publishing your paper in Nature Methods, but would like to consider your response to these concerns before we reach a final decision on publication.

We therefore invite you to revise your manuscript to address these concerns. We would like to ask for an additional demonstration of m6Anet in a different organism, as also requested by the reviewers. We think reviewer#3's concern regarding the confounded training data is critical and should be fully addressed.

[Redacted] This URL links to your confidential home page and associated information about manuscripts you may have submitted, or that you are reviewing for us. If you wish to forward this email to co-authors, please delete the link to your homepage.

We hope to receive your revised paper within 10 weeks. If you cannot send it within this time, please let us know. In this event, we will still be happy to reconsider your paper at a later date so long as nothing similar has been accepted for publication at Nature Methods or published elsewhere.

OPEN SCIENCE REQUIREMENTS

REPORTING SUMMARY AND EDITORIAL POLICY CHECKLISTS

IMAGE INTEGRITY

DATA AVAILABILITY

We strongly encourage you to deposit all new data associated with the paper in a persistent repository where they can be freely and enduringly accessed. We recommend submitting the data to discipline-specific and community-recognized repositories; a list of repositories is provided here:

<http://www.nature.com/sdata/policies/repositories>

All novel DNA and RNA sequencing data, protein sequences, genetic polymorphisms, linked genotype and phenotype data, gene expression data, macromolecular structures, and proteomics data must be deposited in a publicly accessible database, and accession codes and associated hyperlinks must be provided in the “Data Availability” section.

Please include a “Data availability” subsection in the Online Methods. This section should inform readers about the availability of the data used to support the conclusions of your study, including accession codes to public repositories, references to source data that may be published alongside the paper, unique identifiers such as URLs to data repository entries, or data set DOIs, and any other statement about data availability. At a minimum, you should include the following statement: “The data that support the findings of this study are available from the corresponding author upon request”, describing which data is available upon request and mentioning any restrictions on availability. If DOIs are provided, please include these in the Reference list (authors, title, publisher (repository name), identifier, year). For more guidance on how to write this section please see:

<http://www.nature.com/authors/policies/data/data-availability-statements-data-citations.pdf>

CODE AVAILABILITY

Please include a “Code Availability” subsection in the Online Methods which details how your custom code is made available. Only in rare cases (where code is not central to the main conclusions of the paper) is the statement “available upon request” allowed (and reasons should be specified).

MATERIALS AVAILABILITY

SUPPLEMENTARY PROTOCOL

To help facilitate reproducibility and uptake of your method, we ask you to prepare a step-by-step Supplementary Protocol for the method described in this paper. We [encourage authors to share their step-by-step experimental protocols](https://www.nature.com/nature-research/editorial-policies/reporting-standards#protocols) on a protocol sharing platform of their choice and report the protocol DOI in the reference list. Nature Research's Protocol Exchange is a free-to-use and open resource for protocols; protocols deposited in Protocol Exchange are citable and can be linked from the published article. More details can found at www.nature.com/protocolexchange/about.

ORCID

Nature Methods is committed to improving transparency in authorship. As part of our efforts in this direction, we are now requesting that all authors identified as ‘corresponding author’ on published papers create and link their Open Researcher and Contributor Identifier (ORCID) with their account on the Manuscript Tracking System (MTS), prior to acceptance. This applies to primary research papers only. ORCID helps the scientific community achieve unambiguous attribution of all scholarly

contributions. You can create and link your ORCID from the home page of the MTS by clicking on 'Modify my Springer Nature account'. For more information please visit www.springernature.com/orcid.

Best regards,
Lei

Lei Tang, Ph.D.
Senior Editor
Nature Methods

Reviewers' Comments:

Reviewer #1:

Remarks to the Author:

In this manuscript Henra et al. introduce m6Anet, a supervised machine learning algorithm for predicting m6A methylation in nanopore direct RNA sequencing data. m6Anet uses Multiple Instance Learning (MIL) to call the methylation status of each site on a per-read level, and then pools the results into per-site methylation probabilities, which the authors show corresponds to the underlying stoichiometry. They train and test m6Anet on human HCT116 and HEK293T cell lines using labels from m6ACE-Seq data, and show that the results remain accurate when trained and tested using different genes from different cell lines. They compare their results against Tombo, EpiNano, nanom6A, and MINES, and show that m6Anet is substantially more accurate than all these methods. Comparing to m6ACE-Seq and miCLIP data, two immunoprecipitation-based methylation detection methods, they find that few sites are identified as methylation by all three methods, however m6Anet shares more sites between each method than the other methods share with each other. They also show that the methylation patterns across transcript regions are consistent with other methods, and that m6Anet can detect changes in methylation status caused by METTL3 knockouts. Finally, they show that m6Anet can detect changes in methylation stoichiometry by mixing different concentrations on wild-type and knockout samples and showing that the methylation probabilities correspond to the expected methylation fraction.

The results in this manuscript are very promising and substantially outperform all methods they compared against. The code was also very easy to install and run on my own computer. The main

technical development in this method is the MIL framework used to call methylation on a per-read level. While I'm not aware of another project which uses this exact framework, the authors seem to have missed a recent RNA modification detection tool which detects modifications on the per-read level: nanoRMS (Begik et al., Quantitative profiling of pseudouridylation dynamics in native RNAs with nanopore sequencing, Nature Biotechnology 2021). NanoRMS was designed to detect pseudouridine, but like m6Anet, the authors state that it could be applied to any modification type. The methods are quite different (nanoRMS is not neural network based), but they do output per-read methylation calls and estimate site-level stoichiometry. Re-training nanoRMS for m6A modifications may not be practical, but some discussion of the similarities and differences between these methods should be included.

By training and testing on different cell lines they demonstrated that m6Anet does not appear to be overtrained on any specific human sample, although since it was only trained with human data it is not clear if it would generalize to other species or even highly aberrant human cell lines. In particular, Fig. 1C shows the very uneven distribution of methylated k-mers in the data, and I wonder how much this could vary between species/samples. There is at least one project which examined Arabidopsis m6A methylation with publically nanopore sequencing data (Parker et al., Nanopore direct RNA sequencing maps the complexity of Arabidopsis mRNA processing and m6A modification, eLife 2020), which should be used to demonstrate greater generalizability (or a similar dataset, if available).

Minor comments:

- When describing ONT RNA-Seq in the third paragraph of the introduction, they write "...when an oligonucleotide passes through the pore...". Is this limited to oligonucleotides, or any kind of RNA molecule?
- The Fig. 1b axis labels and caption should clarify that they are plotting normalized units. It is described as showing the "difference in average features distribution", which implies subtraction was used. "Comparison" might be a more accurate description.
- The Fig. 1c caption says it is plotting "the top 4 modified 5-mers", while the actual figure displays 18 5-mers
- The Fig. 2e caption says it is plotting the "true positive rate", while the figure axis is labeled "precision". Please clarify

Reviewer #2:

Remarks to the Author:

In this paper, “Detection of m6A from direct RNA sequencing using a multiple Instance Learning framework,” Hendra et al. summarize their work on “m6Anet,” a neural network-based method that uses multiple instance learning framework to handle missing read-level modification labels in site-level training data. They claim that m6Anet outperforms existing computational methods, shows similar accuracy as experimental approaches, and generalizes to different cell lines. This paper is a welcome additional tool to the field, but a few other questions remain about the paper and its data:

- 1) The authors performed 5-fold cross validation with bases features (0-5 bases) flanking the candidate sites to evaluate the additional value of neighboring positions, but were any of these predictions affected by secondary structure?
- 2) Figures 2a and 3a/b show a Venn diagram, but the area shown in each region should be proportional to the number of sites (a proportional Venn).
- 3) The authors state that the “precision of m6Anet is underestimated when comparing it to labels obtained from miCLIP or m6ACE-Seq, most likely reflecting technology-specific m6A predictions,” but the best way to know this would be with a synthesized RNA molecule; have the authors looked at in vitro created RNA molecules that harbor the exact number of sites?
- 4) The authors used a “hyper-dimensional representation of each read based on its signal and sequence features” to infer a read-level modification probability, but their data (Figure 4) shows a lot of overlap between these data clouds. Are the reads that are more ambiguous defined by any features that could separate them in a better way? For example, GC composition, secondary structure, or known motif densities?
- 5) The m6A stoichiometry data on the METTL3 knockout and wild type samples is among the most compelling data, but it is not clear if the proportionality was equally effective across all genes or transcript types. An analysis of this by gene and transcript type (e.g. nuclear-enriched vs. cytosolic, or ribosome-associated vs. not) would be interesting and may explain some of the observed variance.
- 6) Have the authors observed similar success with yeast or other direct RNA data? Their model might be overly-trained on the human data here, and it would be good to see validation on another model.
- 7) Oxford Nanopore has recently released Q20 chemistry on their platform, and it would be good to address this as well in the paper, if possible. Some of this is referenced in the discussion, but we have seen dramatic differences in motif calling and base quality as a function of the version of the base caller, and this will likely be an issue for m6Anet; an estimate of this impact would be helpful for users of the software.

Reviewer #3:

Remarks to the Author:

This manuscript describes m6Anet, a framework for detecting RNA modifications from nanopore direct RNA sequencing data using multiple instance learning. The authors have demonstrated the framework by training with m6ACE-seq data to detect N(6)methyladenosine. There are now many methods for detecting m6A in nanopore data, including another by the authors themselves (xPore). The main advantage of the m6Anet framework over these previous methods is that it does not require a low

modification control. This constitutes a significant advance. However, the manuscript is heavily focused on human data and more work is needed to demonstrate that their models can generalise to datasets from other organisms.

- The data used to train the model seems likely to be partially confounded by sequence context because the authors use all DRACH motifs as negative examples. In reality, some DRACH motifs are much more commonly methylated than others (for example, there are ~1500 times more GGACU motifs than UAACC motifs in the authors positive training examples) whereas the distribution of motifs for negative samples is much more uniform (e.g. only 1.7 times more negative GGACU examples than UAACC). This means that the model can achieve a good accuracy on the training data by learning the motif bias of m6A, rather than by identifying methylation from signal data. When applied to a dataset where the m6A motif preference deviates from this expectation, i.e. in other organisms besides humans, m6Anet may perform sub optimally. Using the authors' training data (file: `data/cv_results/1_neighbour/test_results_pr_auc.csv.gz`) I find that an extremely random forest classifier trained on one hot encoding of the central 5mer sequence (with random oversampling of positive examples) can achieve a 5-fold cross-val AUC of 0.80. The authors should therefore undersample their negative training examples to make the kmer distribution more similar to the positive examples. Alternatively, they could try training a model using signals from their METTL3 KO data at positions matched with positive examples from untreated data, so that sequence contexts are identical between positive and negative examples.
- The authors train their model on modified and unmodified DRACH kmers from HCT116 cells, and then test on modified and unmodified DRACH kmers from HEK293T cells. Given that many positions between these cells will have identical contexts (presumably the authors used the same reference genome/transcriptome) this could be considered at risk of data leakage, since overfitting to the training data could provide a better score on the test set. The authors should alleviate this concern by benchmarking on a held-out set from HCT116 cells or using cross-validation scores to benchmark their model against others.
- It is interesting that m6Anet performs much better on HCT116 cells than HEK293T cells, even when the model is trained on HEK cells. The difference in ROC AUC score for example is quite large (~0.84 for HEK cells vs ~0.93 for HCT cells). Can the authors shed any light on why this might be occurring?
- The authors suggest that their model is able to generalise by demonstrating its use on datasets from other human cell lines. However, given that the sequence composition and m6A motif preference of these cell lines will be very similar to the HCT116 data used in training, I do not think that this demonstrates the level of generalisation that users of m6Anet would likely desire. The authors should therefore demonstrate that m6Anet generalises to other species with known differences in their m6A profile. For example, there is publicly available nanopore data for *Plasmodium falciparum* (Lee et al., 2021), *Arabidopsis thaliana* (Parker et al., 2020), and *Toxoplasma gondii* (Farhat et al., 2021; Lee et al., 2021), which have a stronger preference for A at -1 and -2 positions from m6A compared to humans (Baumgarten et al., 2019; Parker et al., 2020). These are also direct RNA datasets available for *Mus musculus* (Sessegolo et al., 2019) and *C. elegans* (Roach et al., 2020), the latter of which has mRNA m6A but not in DRACH contexts (only METTL16 is conserved; Mendel et al., 2021).

- m6Anet relies heavily on nanopolish to provide event level data (mean, std and dwell) for each transcriptomic position. This should be mentioned and cited in the results section “Training data for m6Anet model parameter estimation”.
- Font size on figures is very small throughout, and should be increased.
- Figure 1c, error bars are hard to read (some are also missing or misaligned) and are not described in the figure legend. I think they have been generated with seaborn barplot so are probably bootstrapped 95% confidence intervals (of per gene kmer mod rate?). Legend also states it is only top 4 kmers but all are shown.
- Fig 2d, m6ACE-seq is labelled in legend as orange but there is no orange line. Supp fig 2b, background is labelled in legend as grey but there is no grey line.
- Figure 3e histogram bins are not aligned.

References:

- Baumgarten S, Bryant JM, Sinha A, Reysen T, Preiser PR, Dedon PC, Scherf A. 2019. Transcriptome-wide dynamics of extensive m6A mRNA methylation during Plasmodium falciparum blood-stage development. *Nat Microbiol* 4:2246–2259.
- Farhat DC, Bowler MW, Communie G, Pontier D, Belmudes L, Mas C, Corrao C, Couté Y, Bougdour A, Lagrange T, Hakimi M-A, Swale C. 2021. A plant-like mechanism coupling m6A reading to polyadenylation safeguards transcriptome integrity and developmental gene partitioning in Toxoplasma. *Elife* 10. doi:10.7554/eLife.68312
- Lee VV, Judd LM, Jex AR, Holt KE, Tonkin CJ, Ralph SA. 2021. Direct Nanopore Sequencing of mRNA Reveals Landscape of Transcript Isoforms in Apicomplexan Parasites. *mSystems* 6. doi:10.1128/mSystems.01081-20
- Mendel M, Delaney K, Pandey RR, Chen K-M, Wenda JM, Vågbø CB, Steiner FA, Homolka D, Pillai RS. 2021. Splice site m6A methylation prevents binding of U2AF35 to inhibit RNA splicing. *Cell* 184:3125-3142.e25.
- Parker MT, Knop K, Sherwood AV, Schurch NJ, Mackinnon K, Gould PD, Hall AJ, Barton GJ, Simpson GG. 2020. Nanopore direct RNA sequencing maps the complexity of Arabidopsis mRNA processing and m6A modification. *Elife* 9. doi:10.7554/eLife.49658
- Roach NP, Sadowski N, Alessi AF, Timp W, Taylor J, Kim JK. 2020. The full-length transcriptome of *C. elegans* using direct RNA sequencing. *Genome Res* 30:299–312.
- Sessegolo C, Cruaud C, Da Silva C, Cologne A, Dubarry M, Derrien T, Lacroix V, Aury J-M. 2019. Transcriptome profiling of mouse samples using nanopore sequencing of cDNA and RNA molecules. *Sci Rep* 9:14908.

Reviewer #4:

Remarks to the Author:

Oxford Nanopore Technologies Nanopore sequencing platform remains the only commercially available sequencing platform that directly measures single RNA and DNA molecules. Hence, it can provide

information on the base sequence of DNA and RNA molecules and measure distinct chemical modifications of individual bases of said nucleotide sequences.

While the raw nanopore signal is rich in information, reliably extracting specific parameters such as modification status of RNA bases remains a yet not fully resolved challenge in the field of machine learning. More conventional machine learning methods such as Hidden Markov Models for base and base modification calls have now been replaced by applying deep neural network models. This methodological approach critically relies on extensive training datasets with a known "ground truth". Unfortunately, such datasets currently cannot be generated for all base modifications of interest with the precision required for most deep learning methods, neither with biological nor molecular protocols.

Hendra, Göke, and colleagues address this relevant challenge in their manuscript "Detection of m6A from direct RNA sequencing using a Multiple Instance Learning (MIL) framework by implementing MIL for calling 6mA-modification in nanopore direct RNA sequencing datasets.

The authors provide several rationally designed experiments and analyses corroborating the assumption that the performance of their novel approach surpasses those of existing tools and may be a first-in-class tool enabling RNA 6mA modification calling on the single-molecule level using the MIL approach.

Overall, we find m6ANet to be a promising tool for detecting RNAs methylated at 6A using native nanopore sequencing and will recommend it for publication. The use of the Multiple Instance Learning model is novel and scientifically sound, and it is exciting to see more use of a more comprehensive array of neural network methods in computational biology. Additionally, the manuscript includes several convincing analyses that their results are concordant with orthogonal (experimental) approaches. Finally, we were easily able to reproduce the results from the paper on CodeOcean as well as easily install the tool on our own servers and run it on the sample data provided in the documentation.

Our criticisms focus mainly on the section entitled Novel m6ANet predictions are sensitive to METTL3 knockout. In the first paragraph, the authors argue that the novel methylation sites predicted by m6ANet are often separately supported by other methylation detection methods and are therefore likely truly methylated. This is not convincing logic as a tool with a very high false-positive rate will, of course, have overlaps with false positives of other methods.

It should also be further clarified that xPore calls differentially methylated sites and is not a "comparative" method to m6ANet. Overall, the take-home message of this paragraph is unclear and ambiguous. A user of your tool would like to know what percentage of the calls made by m6ANet are true positives. However, the authors only explain that m6ANet can reliably detect 46% of KO-sensitive methylation sites. Are these 1888 sites used only novel sites predicted by m6ANet? Can you give a new estimate of the precision of m6ANet if these novel sites are 46% true positives?

Other minor criticisms include:

- (Supp) Figure 2c has "Precision" as the y axis label but then describes the true positive rate in the legend. This is done twice
- Figure 2b has a mismatched color legend

Author Rebuttal to Initial comments

Response to Reviewers

We would like to thank all reviewers for their helpful suggestions and comments. Following these reviews, we have included several additional analyses, and we have made the related changes in the text (highlighted in red). We would like to specifically highlight, that we have changed the order of Figure 2 (revised manuscript: generalizability of m6Anet, now including the cross-species comparison) and Figure 3 (revised manuscript: comparison of different technologies and improved estimation of precision which now includes the analysis of synthetic sequences). While this change improves the logical flow with the new analyses that were suggested, we acknowledge that this might possibly lead to confusion during the evaluation of this revision. To simplify the evaluation of this revision, we have included all the updated figures that are relevant to each specific comment in this response document as well (labelled as Figures R1.1 etc). We again would like to thank the reviewers for their constructive comments, which we hope have led to a substantially improved manuscript. Please find our detailed responses below.

Reviewers' Comments:

Reviewer #1:

Remarks to the Author:

In this manuscript Henra et al. introduce m6Anet, a supervised machine learning algorithm for predicting m6A methylation in nanopore direct RNA sequencing data. m6Anet uses Multiple Instance Learning (MIL) to call the methylation status of each site on a per-read level, and then pools the results into per-site methylation probabilities, which the authors show corresponds to the underlying stoichiometry. They train and test m6Anet on human HCT116 and HEK293T cell lines using labels from m6ACE-Seq data, and show that the results remain accurate when trained and tested using different genes from different cell lines. They compare their results against Tombo, EpiNano, nanom6A, and MINES, and show that m6Anet is substantially more accurate than all these methods. Comparing to m6ACE-Seq and miCLIP data, two immunoprecipitation-based methylation detection methods, they find that few sites are identified as methylation by all three methods, however m6Anet shares more sites between each method than the other methods share with each other. They also show that the methylation patterns across transcript regions are consistent with other methods, and that m6Anet can detect changes in methylation status caused by METTL3 knockouts. Finally, they show that m6Anet can

detect changes in methylation stoichiometry by mixing different concentrations on wild-type and knockout samples and showing that the methylation probabilities correspond to the expected methylation fraction.

Response:

We thank Reviewer #1 for their very positive feedback to our manuscript. In their detailed comments, Reviewer #1 has pointed out several inaccuracies in our manuscript figures and sentences which we have corrected accordingly. Furthermore, Reviewer #1 raised an important issue regarding the cross-species generalisability of m6Anet and recommended additional validation on a Arabidopsis dataset ¹ which we have now included in this response as well as in the revised manuscript. We thank Reviewer #1 for their comments, which among others, has greatly improved the section on generalizability. Please find our detailed response below.

Reviewer #1 (Point 1):

The results in this manuscript are very promising and substantially outperform all methods they compared against. The code was also very easy to install and run on my own computer. The main technical development in this method is the MIL framework used to call methylation on a per-read level. While I'm not aware of another project which uses this exact framework, the authors seem to have missed a recent RNA modification detection tool which detects modifications on the per-read level: nanoRMS (Begik et al., Quantitative profiling of pseudouridylation dynamics in native RNAs with nanopore sequencing, Nature Biotechnology 2021). NanoRMS was designed to detect pseudouridine, but like m6Anet, the authors state that it could be applied to any modification type. The methods are quite different (nanoRMS is not neural network based), but they do output per-read methylation calls and estimate site-level stoichiometry. Re-training nanoRMS for m6A modifications may not be practical, but some discussion of the similarities and differences between these methods should be included.

Response:

We thank Reviewer #1 for these very positive comments on our manuscripts as well as pointing out about the NanoRMS paper that we have missed in the original version of the manuscript. NanoRMS detects pseudouridine modifications, therefore we have not included it in the comparison to m6Anet (even though it could probably be retrained with substantial effort as mentioned by the reviewer). However, NanoRMS is based on EpiNano, which we included in many of the evaluations presented in this manuscript.

NanoRMS is a supervised method for detection of pseudouridine from direct RNA-Seq data. In their manuscript, the authors of NanoRMS also present estimates for stoichiometry predictions. However, NanoRMS requires unmodified control samples, as the stoichiometry prediction is based on an unsupervised approach that is also implemented in NanoRMS. In contrast, m6Anet returns per molecule predictions from a single sample as part of its end-to-end model that inherently estimates the read level modification probabilities.

We agree with Reviewer #1 that the NanoRMS paper is highly relevant for this manuscript. In the revised manuscript we now introduce NanoRMS as a supervised method for detection of non-m6A modifications, and we discuss its ability to infer the modification stoichiometry from direct RNA-Seq data (see Introduction and Discussion).

Reviewer #1 (Point 2):

By training and testing on different cell lines they demonstrated that m6Anet does not appear to be overtrained on any specific human sample, although since it was only trained with human data it is not clear if it would generalize to other species or even highly aberrant human cell lines. In particular, Fig. 1C shows the very uneven distribution of methylated k-mers in the data, and I wonder how much this could vary between species/samples. There is at least one project which examined Arabidopsis m6A methylation with publically nanopore sequencing data (Parker et al., Nanopore direct RNA sequencing maps the complexity of Arabidopsis mRNA processing and m6A modification, eLife 2020), which should be used to demonstrate greater generalizability (or a similar dataset, if available).

Response:

In our original manuscript we have demonstrated that m6anet generalised across different human cell lines (Hek293T and Hct116). In both cell line data sets that we used, the distribution of modified k-mers is largely similar, with GGACT being the most frequently methylated motif. Reviewer #1 suggests to further evaluate the generalizability of m6Anet when a sample is provided where the distribution of modified k-mers differs from the data that was used for training (as is the case for predictions across different species, as suggested by Reviewer #1, as well as as Reviewers #2 and #3). Following this suggestion we have evaluated the generalizability of m6Anet using the Arabidopsis direct RNA-Seq data (Parker et al. 2020) that was recommended by Reviewer #1 as a scenario in which the model is trained and tested on datasets with different frequencies of methylated 5-mers.

The Parker et al data set provides direct RNA-Seq data of a cell line that includes a mutant defective m6A writer complex (*VIRILIZER*, vir-1 cell line), and a VIR complemented cell line that restores VIR activity (vir-1c cell line). In order to obtain training labels for m6Anet, we identified differentially modified sites using xPore (p-value < 0.05 after multiple testing correction) (Pratanwanich et al. 2021). We combined the predictions from xPore with the predictions by Parker et al (using an alternative approach described in their manuscript).

Together, we obtain 5950 m6A sites, which we then split into independent training and test data using the same strategy as described in our manuscript (Figure R1a,b). A comparison of the relative frequency of methylated kmers in the human and arabidopsis data sets confirms that both species show substantially different m6A k-mer preferences (Figure R1.1c). Motifs such as GGACT, GAACT, and GGACA that are the most prevalently methylated in the human cell lines make up a significantly smaller proportion of methylated motifs in the Arabidopsis vir-1c cell line. In contrast, the most frequently methylated kmers in Arabidopsis are AA ACT, AGACT, and AGACA, which are less frequently in the human m6A data, suggesting a significant shift in the distribution of methylated motifs between the two datasets (Figure R1c).

To evaluate the generalizability of m6Anet to systems or species that use distinct profiles of methylated k-mer frequencies we trained m6Anet on these data, and evaluated the performance on the human cell line data used in our original manuscript (Figure R1d, e).

Despite the shift in the relative frequency of the methylated motifs, we observe a comparable performance between the models trained on human cell lines and the model trained on the Arabidopsis dataset (Figure. R1.1d-e). Furthermore, the predicted m6A sites display a strong 3'UTR enrichment that is typical for m6A and which is similar to the human trained models (Figure R1.1d,e). In contrast to the human training data which is based on m6ACE-Seq and miCLIP, the Arabidopsis training data used m6A sites identified from direct RNA-Seq data with xPore. Incorporating the predicted m6A sites from xPore in human cell line (METTL3 knockout sensitive positions) in the evaluation improves the precision of the top positions predicted by the Arabidopsis model, matching those of the models trained on the human cell lines (Figure R1.1f). These results suggest that the true precision of the Arabidopsis m6Anet models might be underestimated since it captures sites detected by comparative, direct RNA-Seq based methods that are not always captured by miCLIP or m6ACE, as reported in our original manuscript (Suppl. Figure 3d). Noteworthy, the performance of m6Anet trained on Arabidopsis and tested on human cell lines is still better than existing approaches such as EpiNano, even though these were trained and tested on the same species (human cell line data) (Fig. 1d-f and Supplementary Fig.1s-u).

Next, we evaluated the prediction made by m6Anet against the m6A sites in

Arabidopsis, which confirm our findings on the human cell lines that m6Anet generalises well when other methylated k-mer frequencies are observed (Figure R1.2a,b,c). Using the METTL3-sensitive sites identified by xPore in human cell lines for training improves the performance, achieving a similar level of precision as the Arabidopsis trained model (Figure R1.2a,b). The predicted m6A sites of all models display a strong 3'UTR enrichment, suggesting that the models indeed capture genuine methylated sites (Figure R1.2c). A comparison of the ranking of k-mer frequencies predicted to have m6A from m6Anet, xPore, and Parker et al shows a high level of agreement, in particular for the frequently modified, high ranking kmers (Figure R1.2d,e,f).

Together, these results demonstrate that the pre-trained models from m6Anet generalise well to other cell lines from the same species, but also to different species that use distinct frequencies of methylated kmers. One limitation of supervised methods such as m6Anet is the requirement for training data: a complete lack of training data for example due to species-specific k-mers will impact the ability to generalise. While training data such as miCLIP is often difficult to obtain, we now demonstrate that m6Anet can also be trained using modification labels purely derived from a comparative analysis of direct RNA-Seq data. While we show that the pre-trained models from m6Anet generalise well, this approach will enable the analysis of m6A (and other modifications) in the scenario when entirely distinct kmers are expected to be modified.

We thank Reviewer #1 for their suggestion, which we believe has led to a greatly improved analysis on generalisability of m6Anet. We have included these new analyses in the main text p. 8 paragraph 2, p. 9 paragraph 1 and in Figure 2, Suppl. Figure 2, Suppl. Figure 3d.

Figure R1.1 Comparison of m6Anet models on human cell lines

(a-b) Distribution of modified positions across all three cell lines on the training sets and the test sets (area shown to proportion) (c) Barplot comparing the relative proportion of methylated motifs for the HCT116, HEK293T and Arabidopsis VIR-1 complemented cell

lines. **(d-i)** ROC curve, PR Curve, and metagene plot of sites predicted by models trained on the HCT116, HEK293T and VIR-1 complemented cell lines on **(d-f)** HCT116 test set and **(g-i)** HEK293T test set. **(j)** The adjusted precision after including position sensitive to METTL3-KO of all three m6ANet models on the HEK293T cell line.

Figure R1.2 Comparison of m6ANet models on Arabidopsis Datasets

(a-b) ROC Curve and PR Curve of four m6ANet models trained on HCT116 cell line, HEK293T cell line, Arabidopsis VIR-1 complemented cell line, and HEK293T cell line with the inclusion of KO sensitive positions as detected by xPore on the Arabidopsis VIR-1 complemented cell line test set **(c)** Metagene plot of predicted sites by all four m6ANet models **(d)** Scatter plot comparing the frequency ranking of predicted motifs by m6ANet against Parker et al and **(e)** xPore and **(f)** xPore against Parker et al

Reviewer #1:

Minor comments:

- When describing ONT RNA-Seq in the third paragraph of the introduction, they write "...when an oligonucleotide passes through the pore...". Is this limited to oligonucleotides, or any kind of RNA molecule?

Response:

We thank Reviewer #1 for pointing out this sentence. Nanopore RNA-Seq is not limited to oligonucleotides, we have therefore modified the sentence accordingly:

Introduction, p.3:

*“The ability to sequence native RNA using Oxford Nanopore direct RNA-Seq can potentially overcome these limitations ². Nanopore direct RNA-Seq infers the RNA sequence using the current intensity when **RNA molecules** pass through the pores.”*

Reviewer #1:

- The Fig. 1b axis labels and caption should clarify that they are plotting normalized units. It is described as showing the “difference in average features distribution”, which implies subtraction was used. “Comparison” might be a more accurate description.
- The Fig. 1c caption says it is plotting “the top 4 modified 5-mers”, while the actual figure displays 18 5-mers

Response:

Following the suggestion, we have revised the caption and axis labels of Figure 1b to explain that the boxplots show the features in their normalised units. Reviewer #1 also rightly pointed out that our description of the boxplots showing “[...]difference in average features distribution” can be misleading and have therefore followed the reviewer’s suggestion to revise the caption to “[...]comparing normalised features distribution.” Lastly, we thank Reviewer #1 for finding the error in the caption of Fig. 1c, which we have corrected in the revised manuscript (it should be 18 5-mers instead of top 4 modified 5-mers). We thank the Reviewer for this comment.

Reviewer #1:

- The Fig. 2e caption says it is plotting the “true positive rate”, while the figure axis is labeled “precision”. Please clarify

Response:

We thank Reviewer #1 for finding this error in Fig. 2e (it should be precision). We have corrected the figure caption in the revised manuscript.

Reviewer #2:

Remarks to the Author:

In this paper, “Detection of m6A from direct RNA sequencing using a multiple Instance Learning framework,” Hendra et al. summarize their work on “m6Anet,” a neural network-based method that uses multiple instance learning framework to handle missing read-level modification labels in site-level training data. They claim that m6Anet outperforms existing computational methods, shows similar accuracy as experimental approaches, and generalizes to different cell lines. This paper is a welcome additional tool to the field, but a few other questions remain about the paper and its data:

Response:

We thank Reviewer #2 for their positive comments and constructive suggestions. In particular the suggestion to use synthetic data has led to a greatly improved ability to estimate the precision of m6Anet at the site level and for single molecule predictions. Please find our detailed response to this and all other comments below.

Reviewer #2:

- 1) The authors performed 5-fold cross validation with bases features (0-5 bases) flanking the candidate sites to evaluate the additional value of neighboring positions, but were any of these predictions affected by secondary structure?

Response:

The m6Anet model uses signal based features (mean, standard deviation, and length) that are affected by the shape of the RNA molecules that go through the nanopore and the time that the specific 6-mer is within the pore. The molecule has to be unstructured to be sequenced by a nanopore, therefore the mean signal level is not expected to be influenced by RNA secondary structures. However, highly structured RNAs can block the pores, leading to reduced throughput, shorter read length, and possibly longer signal length per event, which might affect predictions by m6Anet. To minimise the impact of secondary structure on direct RNA-Seq, the sequencing process involves an optional reverse transcription (RT) step, which adds a cDNA strand to the RNA molecules. The cDNA strand is not sequenced and does not alter the modifications on the RNA strand, however, this step removes intramolecular secondary structure, and thereby increases throughput and average read length (Garalde et al. 2018). With this RT step, the direct RNA-Seq reads are comparable in length to Nanopore cDNA sequencing, indicating that the impact of secondary structure on the sequencing process is largely removed³

Detection of secondary structure is also possible with direct RNA-Seq^{4,5}. In these

studies the authors introduce artificial modifications that specifically target secondary structures so that they can be detected from direct RNA sequencing. Without these artificial modifications, the RNA structures are not detectable, further suggesting that the presence of secondary structure might only minimally impact the signal intensity detected in the pores.

Reviewer #2:

2) Figures 2a and 3a/b show a Venn diagram, but the area shown in each region should be proportional to the number of sites (a proportional Venn).

Response:

In our manuscript we use Venn diagrams to show the different numbers of m6A sites for different technologies (Figure 2a) or for the number of m6A sites used for training and testing (Figure 3a/b). The areas in each region are actually proportional to the number of sites in these figures, however, as the total number of sites shown in each region is approximately equal to each other, this is not immediately obvious to the readers. To clarify this in the revised manuscript, we have modified the figure legend to explicitly state that the areas of the Venn diagrams are proportional to the number of sites (Figure R2.1,a-c corresponding to Figures 2a and 3a,b). Additionally, we have also corrected an error where the labels for the venn diagram for HCT116 and HEK293T were swapped in Fig.3a,b in the original manuscript. We thank #Reviewer 2 for highlighting this point, which we hope is now more clearly described in the revised manuscript.

Figure R2.1. Venn diagram of DRACH m6A sites detected between m6ACE, miCLIP, and m6ANet in HEK293T cell lines. (a) Total number of modified sites captured

by *m6Anet*, m6ACE-seq and miCLIP (area in (a) shown to proportion). Figure corresponds to Figure 2a.

(b) Distribution of modified positions across both cell lines on the training sets and (c) the test sets (area shown to proportion). Figure corresponds to Figure 3a.

Reviewer #2:

3) The authors state that the “precision of m6Anet is underestimated when comparing it to labels obtained from miCLIP or m6ACE-Seq, most likely reflecting technology-specific m6A predictions,” but the best way to know this would be with a synthesized RNA molecule; have the authors looked at in vitro created RNA molecules that harbor the exact number of sites?

Response:

In our original manuscript, we made the claim that the precision of m6Anet might be underestimated due to incomplete m6a labels from both miCLIP or m6ACE-Seq. We validated that claim using direct RNA-Seq data from a METTL3 knockout cell line compared to a wild type cell lines, which indicated that a substantial fraction of sites detected by m6Anet which were not detected by m6ACE-Seq or miCLIP were sensitive to loss of METTL3 (Figure 2e). Reviewer #2 suggests that this claim can additionally be validated using synthetic sequences where m6A labels are complete.

Following this suggestion, we downloaded synthetic sequences (“curlcake sequences”) provided by⁶ that contain two replicates of an m6A modified library and two replicates of an unmodified IVT RNA library. In order to evaluate the accuracy of m6Anet at different levels of modified RNAs, we followed the strategy used by the authors to randomly sample reads from the modified and unmodified libraries to create validation sets containing various percentages of m6A modified reads. Similarly, we also excluded 5-mers with multiple modified A nucleotides since they are unlikely to occur in reality⁶.

Using these data, we applied the m6Anet model trained on the human cell line to predict m6A sites on these synthetic sequences. We observe a near optimal median ROC AUC and Precision-Recall AUC (>0.98) with at least 50% modified reads (Figure R2a,b). As the synthetic data set has well controlled modification rates at each position, it also allows us to study the sensitivity of m6anet in relation to the modification stoichiometry. With modification rates between 25% and 50% m6Anet still achieves highly accurate classification (AUC>0.93). Even at the lowest modification mixture (5% methylation level for all methylated positions), our model achieves a

competitive ROC AUC of 0.65 (Figure R2a,b). A comparison with the results presented by Liu et al (EpiNano) shows that m6Anet achieves higher performance and higher sensitivity to detect sites with low modification rates, despite being trained on a different dataset.

These results suggest that the accuracy of m6anet is indeed significantly higher when all the labels in the datasets are known, confirming our original observation that the evaluation with miCLIP and m6ACE-Seq underestimates the precision of m6Anet. We thank Reviewer #2 for the suggestion to investigate synthetic sequences, which not only confirms our initial observations, but further illustrates the high accuracy when labels are known, and the ability to detect sites with low modification rate. We have included these results in the revised manuscript (Figure 3f,g, text p. 10 paragraph 2, p. 11 paragraph 1)

Figure R2.2 m6Anet Results on synthetic sequences with known modification status (Curlicake Dataset)

Boxplots comparing the ROC AUC (a) and PR AUC (b) of m6Anet on curlicake datasets over different mixtures of methylated reads

Reviewer #2:

- 4) The authors used a “hyper-dimensional representation of each read based on its signal and sequence features” to infer a read-level modification probability, but their data (Figure 4) shows a lot of overlap between these data clouds. Are the reads that are more ambiguous defined by any features that could separate them in a better way? For example, GC composition,

secondary structure, or known motif densities?

Response:

We present several visualisations in our original manuscript to illustrate the read level representation learnt by m6Anet (shown as “data clouds”, or reference maps). As pointed out by Reviewer #2, there is a region of overlap between the reads from the wild type (modified) cell line and the METTL3 knockout (unmodified) cell line, raising the question how effective the features used in m6Anet are in separating modified and unmodified reads.

Two aspects will lead to an overlap in these data clouds without affecting the ability to separate modified and unmodified reads in m6Anet from the read level feature space: (1) the visualisation shows a compressed 2 dimensional view of the original 32 dimensional feature space. The reduced feature space is expected to be less informative and show less separation, however that will not impact the classification performance. (2) We generated reference maps using wild type (modified) and METTL3 knockout (unmodified) samples. Not all reads in the wild type sample are modified, therefore the wild type map is expected to partially overlap with the unmodified map. Furthermore, METTL3 independent m6A sites will still be modified in the knockout sample, leading to an overlap with the wild type (modified) map.

That being said, Reviewer #2 has in their earlier comment suggested exploring synthesised RNA molecules that can provide per-read modification labels, and which we can use to quantify how well the read-level representation in m6Anet separates modified and unmodified reads. Using the synthetic data from Liu et al. 2019 (see response above) we ranked individual reads by the predicted read-level probability (Figure R2.3a,b). For single molecule predictions, m6Anet achieves a ROC AUC of 0.90 and a PR ROC of 0.91, suggesting that m6Anet accurately separates individual modified and unmodified reads (Figure R2.3 a,b). When we generated the reference maps for visualisation in 2 dimensions from the synthetic data we still observed an overlap (Figure R2.3c), confirming that this is largely a limitation in the compressed 2-dimensional visualisation of the original 32-dimensional features rather than a limitation in the ability to estimate per read modification status.

To additionally validate the accuracy of read level probabilities provided by m6Anet, we explored their ability to estimate the proportion of modified reads at each position (the *modification rate*, or *stoichiometry*). In our original manuscript, we have used the site-level probability to estimate the modification rate. However, the high level of accuracy for single molecule predictions with m6Anet suggests that the read level probabilities could be directly used to infer a more accurate modification rate. Using

the single read classification ROC (Figure R2.3a) we selected a threshold t that will provide the greatest difference between true positive and false positive rate. We then used this threshold to compute the modification rate for each position as the number of modified reads per site ($p > t$). On the synthetic sequence mixtures (see also response above), the estimated relative modification rate closely matches the expected modification rate, further confirming that m6Anet accurately discriminates between modified and unmodified reads (Figure R2.3,d). Next we tested this on the METTL3 knockout - wild type mixture datasets, where we estimated the relative modification rate at sites that m6Anet detected to be modified in the wild type cell line and which are unmodified in the knockout cell line. Here we found that the median modification rates also match what we expect from each mixture (Fig R.2.3e). Compared to the original estimate of the modification stoichiometry we observe less variation and a closer match to the expected methylation rates, suggesting that the read level predictions improve the stoichiometry estimate over what we originally presented.

Reviewer #2 suggests exploring additional, sequence-based features to improve the separation of reads in the feature space representation. In principle, the addition of more features might help, however, the overlap between data points in the 2-dimensional representation is also observed for reads aligned to the same position when the sequence is identical (e.g. Figure 4c). Since m6Anet aims to separate modified and unmodified reads for the same position rather than across different positions, additional sequence-specific features such as secondary structure, GC composition and motif density that will be identical for the same position are unlikely to increase the ability to separate reads beyond the high accuracy that we already observe on the synthetic data.

We thank Reviewer #2 for this comment and the suggestion to use the synthetic data, which has allowed us to quantify the accuracy to separate modified and unmodified reads and which has led to an improved estimation of the modification stoichiometry. In our revised manuscript we now include these additional analyses that complement the visualisation of single molecule predictions in the main text p. 11 paragraph 2, p.12 paragraph 1-3, p.13 paragraph 1 and in Figures 4a-c, g,h.

Figure R2.3 m6Anet Results on Individual Read Methylation Prediction

(a) ROC Curve and PR Curve of m6Anet single molecule prediction on IVT and synthesised RNA reads from curlicake datasets (c) Density plot of PCA projected read features for both modified and unmodified (IVT) RNA reads from the curlicake dataset. Box plots comparing the ratio of the predicted modification stoichiometry between the (d) curlicake reads (e) HEK293T cell line with different levels of KO mixture

Reviewer #2:

5) The m6A stoichiometry data on the METTL3 knockout and wild type samples is among the most compelling data, but it is not clear if the proportionality was equally effective across all genes or transcript types. An analysis of this by gene and transcript type (e.g. nuclear-enriched vs. cytosolic, or ribosome-associated vs. not) would be interesting and may explain some of the observed variance.

Response:

We thank Reviewer #2 for this positive comment regarding the ability to estimate the

stoichiometry of m6A. Following the suggestion from Reviewer #2 to explore synthetic sequences we have updated the RNA modification stoichiometry estimate in m6anet, which now achieves even better accuracy with less variation (see response above, Figure R2.3d,e). While the estimates are highly accurate for synthetic sequences, we still observe higher variation in human cell line data (as referred to by Reviewer #2 in this comment). This variation is partially attributed to the data generation procedure: the mixtures were obtained by combining RNA extract from wild type HEK293T cells with METTL3 knockout HEK293T cells. While the average relative modification rate is expected to match the mixture ratio, this is not the case for each individual position/transcript which can't be controlled by mixing total RNA.

In addition, factors such as the gene biotype can also contribute to the observed variation in biological samples. Following the suggestion from Reviewer #2, we have split genes into Nucleus, Cytosol, and Ribosome-associated using the RNALocate database ⁷. In order to compare more transcripts, we lower the threshold for both modified sites and unmodified sites and consider sites with m6Anet predicted probability greater than 0.7 in the WT samples as modified, and lower than 0.3 in the 100% KO samples as unmodified.

Afterwards, we compared the stoichiometry estimates across the mixtures and found that the estimates are highly comparable (Figure R2.4b-e). While some biological factors possibly influence these results, we believe that the observed variation is likely explained by limitations in the RNA mixing procedure.

Figure R2.4 m6Anet Stoichiometry Prediction on curlcake and HEK293T Mixture Datasets

Box plots comparing the ratio of the predicted modification stoichiometry between the (a) curlcake reads (b) HEK293T cell line with different levels of KO mixtures (c-e) HEK293T KO mixtures on transcripts localised to (c) Nucleus (d) Ribosome (e) Cytosol. The x-axis indicates the percentage of WT reads in the mixture while the vertical lines indicate the expected stoichiometric ratio for each mixture with the matching colour.

Reviewer #2:

- 6) Have the authors observed similar success with yeast or other direct RNA data? Their model might be overly-trained on the human data here, and it would be good to see validation on another model.

Response:

In our original manuscript we have validated the generalisability of m6Anet by training and testing on two different human cell lines. One possible limitation with our validation method is the fact that both of these human cell lines contain similar frequencies of methylated 5-mer motifs. Reviewer #2 (as well as Reviewer #1 and Reviewer #3), have

therefore suggested to evaluate the generalisability of m6Anet using data from different species, that harbour different frequencies of methylated kmers.

Following a suggestion from Reviewer #1, we have downloaded a data set from Arabidopsis (Parker et al. 2020) which contains a different distribution of methylated motifs (Fig.

R2.5a,b,c). Motifs such as GGACT, GAACT, and GGACA that dominate the methylated positions in the human cell lines are less prominent in the Arabidopsis dataset. On the other hand, motifs such as AA ACT, AGACT, and AGACA, which are less frequently found in the human m6A data, are more prevalent in the Arabidopsis dataset, suggesting a significant shift in the distribution of methylated motifs between the two datasets, which makes this an ideal data set to test the generalizability of m6Anet beyond the application on human cell lines.

Using these data, we performed two new analysis to test the generalizability of m6anet:

(1) Cross-species prediction (Training: Arabidopsis - Testing: Human)

Here we observe a comparable performance between models trained on human cell lines against the model trained on Arabidopsis dataset (Fig. R2.5d,e). We also note that the model trained on the Arabidopsis cell line also outperforms existing approaches such as EpiNano on both the HCT116 cell line and the HEK293T cell line (Fig. 1d,f and Supplementary Fig. 1s,u). All predicted sites display strong enrichment towards the 3'UTR, indicating that these are genuine m6A sites (Figure R2.5f,i, Figure R2.6c). Including the positions sensitive to knockout in the HEK293T cell line shows an improvement in the precision of the Arabidopsis trained model, matching those of the models trained on the human cell lines (Fig. R1.1j) suggesting a comparable performance and high level of generalisability of m6Anet.

(2) Cross-species prediction (Training: Human - Testing: Arabidopsis)

Similarly, we also compared the performance of m6Anet trained on human cell lines against the arabidopsis model and found comparable performance despite the difference in the distribution of the methylated motifs (Figure. R2.6a,b). Furthermore, incorporating METTL3-sensitive sites detected by xPore in the HEK293T cell line improves the training performance of the model (Fig. R2.6a,b). We have also observed an enrichment around the 3'UTR area for all models (Fig. R2.6c), suggesting that the sites captured by our models are indeed true methylated sites. Lastly, the rankings of the most methylated 5-mer frequencies by m6Anet, xPore, and the results presented by Parker et al show a strong agreement to each other, in particular for the most frequently modified motifs (Figure. R2.6,f,g)

Together these results demonstrate that m6Anet generalises to new species or other samples that have different kmer profiles compared to the training data. We have included these results in the revised manuscript p. 8 paragraph 2, p. 9 paragraph 1 and in Figure 2, Suppl. Figure 2, Suppl. Figure 3d

Figure R2.5 Comparison of m6Anet models on human cell lines

(a-b) Distribution of modified positions across all three cell lines on the training sets and the test sets (area shown to proportion) **(c)** Barplot comparing the relative proportion of methylated motifs for the HCT116, HEK293T and Arabidopsis VIR-1 complemented cell lines. **(d-i)** ROC curve, PR Curve, and metagene plot of sites predicted by models trained on the HCT116, HEK293T and VIR-1 complemented cell lines on **(d-f)** HCT116 test set and **(g-i)** HEK293T test set. **(j)** The adjusted precision after including position sensitive to METTL3-KO of all three m6Anet models on the HEK293T cell line.

Figure R2.6 Comparison of m6Anet models on Arabidopsis Datasets

(a-b) ROC Curve and PR Curve of four m6Anet models trained on HCT116 cell line, HEK293T cell line, Arabidopsis VIR-1 complemented cell line, and HEK293T cell line with the inclusion of KO sensitive positions as detected by xPore on the Arabidopsis VIR-1 complemented cell line test set **(c)** Metagene plot of predicted sites by all four m6Anet models **(d)** Scatter plot comparing the frequency ranking of predicted motifs by m6Anet against Parker et al and **(e)** xPore and **(f)** xPore against Parker et al

Reviewer #2:

7) Oxford Nanopore has recently released Q20 chemistry on their platform, and it would be good to address this as well in the paper, if possible. Some of this is referenced in the discussion, but we have seen dramatic differences in motif calling and base quality as a function of the version of the base caller, and this will likely be an issue for m6Anet; an estimate of this impact would be helpful for users of the software.

Response:

Detection of m6A with m6anet uses the Nanopore current signal, which can potentially be impacted with a new technology release by Oxford Nanopore. We do not have any datasets that are sequenced using the Q20 chemistry (which was released for DNA sequencing, for RNA the old chemistry is still recommended). However, in our original manuscript, we have performed experiments using datasets that are sequenced using different RNA chemistry kits. The HEK293T and HCT116 datasets are sequenced using SQK-RNA001 (HEK293T) and SQK-RNA002 (HCT116) kits and are also basecalled using Guppy version 2.1.3 and 3.2.10. In our manuscript we have demonstrated that m6anet generalises across these two datasets, despite them being sequenced using different RNA kits. Furthermore, we have also included an additional validation result on the Arabidopsis dataset from Parker et al sequenced using the SQK-RNA001 kit and basecalled using Guppy version 2.3.1¹ and showed that our approach generalises to this dataset as well. The generalizability is achieved as we do not observe dramatic differences in the raw squiggle/signal data, which translates to comparable current features across all these datasets. Additionally, the comparable raw squiggle values also ensure consistent segmentation results of the target m6A region by nanopolish eventalign in the squiggle.

Even though m6Anet robustly works across all current RNA chemistry kits from Oxford Nanopore, it is possible that a new pore version or chemistry kit might be released which alters the signal. In that case, the segmentation and signal features are likely to be impacted, requiring re-training of m6anet. m6Anet now includes the option to re-train the model for this scenario. We have now highlighted this fact in paragraph 4 of the discussion section in the manuscript and we further highlighted this in the online documentation of m6anet (<https://m6anet.readthedocs.io/en/latest/>)

Reviewer #3:

Remarks to the Author:

This manuscript describes m6Anet, a framework for detecting RNA modifications from nanopore direct RNA sequencing data using multiple instance learning. The authors have demonstrated the framework by training with m6ACE-seq data to detect N(6)methyladenosine. There are now many methods for detecting m6A in nanopore data, including another by the authors themselves (xPore). The main advantage of the m6Anet framework over these previous methods is that it does not require a low modification control. This constitutes a significant advance. However, the manuscript is heavily focused on human data and more work is needed to demonstrate that their models can generalise to datasets from other organisms.

Response:

We thank Reviewer #3 for their positive evaluation of our manuscript. In our revised manuscript we now demonstrate that m6Anet can accurately predict m6A in a cross-species scenario (Human - Arabidopsis) when the relative frequency of modified k-mers differs substantially. We thank Reviewer #3 for their constructive suggestions, which we believe has led to an improved manuscript that better demonstrated the robustness and generalizability of m6Anet to predict m6A beyond human samples. Please find our detailed responses below.

Reviewer #3 (point1):

- The data used to train the model seems likely to be partially confounded by sequence context because the authors use all DRACH motifs as negative examples. In reality, some DRACH motifs are much more commonly methylated than others (for example, there are ~1500 times more GGACU motifs than UAACC motifs in the authors positive training examples) whereas the distribution of motifs for negative samples is much more uniform (e.g. only 1.7 times more negative GGACU examples than UAACC). This means that the model can achieve a good accuracy on the training data by learning the motif bias of m6A, rather than by identifying methylation from signal data. When applied to a dataset where the m6A motif preference deviates from this expectation, i.e. in other organisms besides humans, m6Anet may perform sub optimally. Using the authors' training data (file: data/cv_results/1_neighbour/test_results_pr_auc.csv.gz) I find that an extremely random forest classifier trained on one hot encoding of the central 5mer sequence (with random oversampling of positive examples) can achieve a 5-fold cross-val AUC of 0.80. The authors should therefore undersample their negative training examples to make the kmer distribution more similar to the positive examples. Alternatively, they could try training a model using signals from their METTL3 KO data at positions matched with positive examples from untreated data, so that sequence contexts are identical

between positive and negative examples.

Response:

Reviewer #3 raises an important point regarding the generalizability of m6Anet to dataset with different methylated motifs distribution due to the imbalance in the methylated DRACH motifs in our dataset (with GGACU motif being much more commonly methylated than the other motifs). Following the suggestion from Reviewer #2, we now present the following additional analysis:

- (1) We now train and test m6Anet on an additional data set from Arabidopsis to demonstrate cross-species generalisability when k-mer profiles differ between training and test data
- (2) We compare the original strategy to train m6Anet (randomly oversampling of methylated positions) with k-mer-specific oversampling of positive labels and

k-mer-specific undersampling of negative labels (as suggested by Reviewer #2) to demonstrate the robustness of the m6Anet pre-trained models

In order to obtain an independent data set to evaluate the generalizability of m6Anet in a scenario where different methylated kmer frequencies are expected, we have followed the suggestion from Reviewer#3 (see below) to use an Arabidopsis data set (Parker et al) (which was also suggested by Reviewers #1 and #2, see response above for additional details).

The Parker et al dataset contains direct RNA-Seq data from a cell line that lacks m6A modifications (vir-1 knockout cell line/ vir-1 cell line) and a matching cell line that contains m6A sites (vir-1 complement,/ vir-1c cell line). To obtain the labels for this dataset, we ran xPore (Pratanwanich et al. 2021) to compare all replicates from the vir-1c dataset against the all four replicates from the vir-1 data set and considered DRACH sites with adjusted

p-value less than 0.05 as m6A modified sites. Additionally, we use the sites provided by the authors as m6A sites for training and testing (see methods). The number of methylated sites (positive labels) in both training and test sets is comparable between the Arabidopsis and the Human data set (Figure.R3.3a,b). However, the relative methylation frequencies between the 5-mers are substantially different (Figure. R3.3,c). In particular, the GGACU motif that is most frequently methylated in the human cell lines is not as prominent in the Arabidopsis dataset. On the other hand, methylated sites that have stronger preference for A at -1 and -2 positions are more frequently modified in Arabidopsis compared to humans (Figure. R3.3c). These data suggest that the Arabidopsis data set provides a good model system to

evaluate the impact of the methylated k-mer frequencies on the generalizability of m6Anet (please see our response below to Reviewer #3 point 4 with additional details about these results and data).

Following the suggestion from Reviewer #3 we compared four different strategies to over- and undersampling training data:

1. The original m6Anet model (“**original**”)
 - a. In the original implementation, we oversample methylated positions to match the number of negative labels. This strategy maximises the amount of data used in m6Anet while maintaining the original relative k-mer frequency.
2. Undersampling of negative labels by kmer (suggested by Reviewer #3) (“**undersampling**”)
 - a. This strategy ensures that the relative k-mer frequency is comparable between the positive and negative labels, at the cost of using less data (in particular for rarely modified k-mers many training data points will not be used)
3. Oversampling of positive labels by k-mers (“**oversampling**”)
 - a. This strategy ensures a comparable relative k-mer frequency between the positive and negative training labels. In contrast to the undersampling strategy, more data points are used.
4. Training on matched wild type and knockout data (using positive label positions only) (suggested by Reviewer #3) (“**matched knockout**”)
 - a. In this strategy we train using only positions identified as modified, with positive labels using wild type cell line data, and negative labels from METTL3 knockout cell line data. This strategy ensures identical sequence context between positive and negative training labels, however it uses a minimum sequence context as none of the unmodified positions is used during training that make the majority of data points)

We applied these strategies to training models from the 2 human cell lines and the additional arabidopsis cell line, and we tested all models on all three data sets (Table R3.1). Firstly, we observe that the models trained by the various sampling strategies do not outperform the model trained on the same cell line using our original strategy, which shows the highest accuracy (Table R.3.1). While the “oversampling” and “undersampling” strategies perform generally well, the “matched knockout” model performs poorly compared to the other models, most likely due to the limited sequence

context the model is trained on compared to all other models. The original strategy as well as the “oversampling” strategy generalise well across species and data sets with very different k-mer profiles (Table R3.1). Overall these results suggest that the original strategy is robust against a possible bias due to the methylated kmer frequency in the training data (Table R3.1). We have included these results in the revised manuscript as supplementary text. For simplicity, we only present the models trained using the original strategy in the manuscript main figures. For more details regarding the cross-species comparison, please see our response below.

HEK293T Test Set								
Training Cell Line	ROC AUC (original)	ROC AUC (Oversampling)	ROC AUC (Undersampling)	ROC AUC (matched knockout)	PR AUC (original)	PR AUC (Oversampling)	PR AUC (Undersampling)	PR AUC (matched knockout)
HEK293T	0.828	0.774	0.664	0.547	0.343	0.271	0.151	0.084
HCT116	0.836	0.793	0.697	NA	0.349	0.280	0.174	NA
Arabidopsis VIRC	0.792	0.793	0.776	NA	0.311	0.314	0.281	NA
HCT116 Test Set								
Training Cell Line	ROC AUC (original)	ROC AUC (Oversampling)	ROC AUC (Undersampling)	ROC AUC (matched knockout)	PR AUC (original)	PR AUC (Oversampling)	PR AUC (Undersampling)	PR AUC (matched knockout)
HEK293T	0.903	0.859	0.775	0.582	0.466	0.327	0.213	0.079
HCT116	0.926	0.898	0.815	NA	0.498	0.380	0.278	NA
Arabidopsis VIRC	0.875	0.874	0.879	NA	0.383	0.389	0.387	NA
Arabidopsis VIRC Test Set								
Training Cell Line	ROC AUC (original)	ROC AUC (Oversampling)	ROC AUC (Undersampling)	ROC AUC (matched knockout)	PR AUC (original)	PR AUC (Oversampling)	PR AUC (Undersampling)	PR AUC (matched knockout)
HEK293T	0.881	0.877	0.896	0.674	0.267	0.249	0.284	0.049
HCT116	0.886	0.898	0.897	NA	0.237	0.238	0.227	NA
Arabidopsis VIRC	0.940	0.937	0.933	NA	0.389	0.346	0.284	NA

Table R3.1 Comparison of mAnet-based models trained on different cell lines with different sampling strategy on the HEK293T Test Set

Reviewer #3 (point 2):

- The authors train their model on modified and unmodified DRACH kmers from HCT116 cells, and then test on modified and unmodified DRACH kmers from HEK293T cells. Given that many positions between these cells will have identical contexts (presumably the authors used the same reference genome/transcriptome) this could be considered at risk of data leakage, since overfitting to the training data could provide a better score on the test set. The authors should alleviate this concern by benchmarking on a held-out set from HCT116 cells or using cross-validation scores to benchmark their model against others.

Response:

In order to address the Reviewer's concern about data leakage, we have validated the model on the HCT116 test set that was not used in training (Fig. R3.2a,b,c). Furthermore, we have now restricted the model comparison between m6Anet and existing approaches to the test set of the HEK293T cell line. The test sets of both HCT116 and HEK293T cell lines do not contain genes that are present in the training dataset and therefore contain different sequence contexts (Figure 1d,e,f). We still observe the same results in which m6Anet outperforms existing approaches (Fig. R3.1a,b,c, Fig. R3.2a,b,c). We thank Reviewer #3 for their suggestions and we have updated the original manuscript (Figure. 1d,e,f, Supplementary Figure. 2j-n)

Figure R3.1 Comparison of *m6Anet* against EpiNano, *nanom6A*, *MINES*, and Tombo on HEK293T test set

(a) ROC Curve and PR Curve of *m6Anet* against all 5 EpiNano models and Tombo. (b) ROC Curve and PR Curve of *m6Anet* against *nanom6A* and Tombo (c) ROC Curve and PR Curve of *m6Anet* against *MINES* and Tombo on the HEK293T test set

Figure R3.2 Comparison of m6Anet against EpiNano, nanom6A, MINES, and Tombo on HCT116 test set

(a) ROC Curve and PR Curve of m6Anet against all 5 EpiNano models and Tombo. (b) ROC Curve and PR Curve of m6Anet against nanom6A and Tombo (c) ROC Curve and PR Curve of m6Anet against MINES and Tombo on the HCT116 test set

Reviewer #3 (point 3):

- It is interesting that m6Anet performs much better on HCT116 cells than HEK293T cells, even when the model is trained on HEK cells. The difference in ROC AUC score for example is quite large (~0.84 for HEK cells vs ~0.93 for HCT cells). Can the authors shed any light on why this might be occurring?

Response:

The models that are trained on the HEK293T and the HCT116 cell lines perform equally on all three data sets used for evaluation in our manuscript (Figure 3c,d), suggesting that the data used in training is of comparable quality in both cell lines (positive labels + direct RNA-Seq data). In contrast, all m6Anet models as well as all other methods (nanom6A, MINES, EpiNano and Tombo) show lower precision when evaluated on the HEK293T data set compared to the HCT116 data set. We believe that this points to a higher number of missing m6A labels in the HEK293T cell line, as these have a much larger impact on evaluation than on model training. When we additionally use the labels obtained from the comparison of the wild type and METTL3 knockout HEK293T cell lines, we can see a significant increase in performance, confirming that a large fraction of m6A positions are likely missing from the labels in the HEK293 data set (Figure R3.3j). The enrichment of the m6A predictions in the 3'UTR is similar between both cell lines (Figure R3.3f,i), further suggesting that the difference in performance between both data sets is more likely to reflect the quality of the labels used during evaluation rather than the quality of the predictions that are made.

Reviewer #3 (point 4):

- The authors suggest that their model is able to generalise by demonstrating its use on datasets from other human cell lines. However, given that the sequence composition and m6A motif preference of these cell lines will be very similar to the HCT116 data used in training, I do not think that this demonstrates the level of generalisation that users of m6Anet would likely desire. The authors should therefore demonstrate that m6Anet generalises to other species with known differences in their m6A profile. For example, there is publicly available nanopore data for *Plasmodium falciparum* (Lee et al., 2021), *Arabidopsis thaliana* (Parker et al., 2020), and *Toxoplasma gondii* (Farhat et al., 2021; Lee et al., 2021), which have a stronger preference for A at -1 and -2 positions from m6A compared to humans (Baumgarten et al., 2019; Parker et al., 2020). These are also direct RNA datasets available for *Mus musculus* (Sessegolo et al., 2019) and *C. elegans* (Roach et al., 2020), the latter of which has mRNA m6A but not in DRACH contexts (only METTL16 is conserved; Mendel et al., 2021).

Response:

In our original manuscript we attempted to validate the generalisability of m6Anet by training and testing on two different human cell lines. Reviewer #3 (as well as Reviewer #1 and Reviewer #2) have pointed out the limitation in our approach due to the similarity in methylated 5-mer composition between the two cell lines. Following the suggestion from the reviewers, we have included an additional validation dataset from

Arabidopsis (Parker et al. 2020) to evaluate the generalizability of m6Anet across different species when different profiles of methylated k-mers are expected (see Reviewer #3 point 1 with a detailed description of the data set).

First we evaluated the m6anet models trained on human cell lines (Hct116, Hek293T) and Arabidopsis cell lines (vir-1) when tested on human data used in our original manuscript. We observe a comparable performance of the m6anet model trained on Arabidopsis data against the other m6Anet models trained on human cell lines when evaluated on the Hek293T cell line (Revision Table R.3.1, Figure R.3.3d,e), and the HCT116 cell line (Revision Table R.3.1, Figure R.3.3.g,h) suggesting that our model can generalise to datasets with different frequency of methylated motifs. Additionally, the metagene plots of all the models on both of these cell lines show strong enrichment towards the 3'UTR indicating that these models capture genuine methylated sites. Incorporating the METTL3-KO sensitive positions obtained from xPore on the HEK293T cell line also improves the precision of the Arabidopsis model, matching that of the models trained on the HEK293T cell line and HCT116 cell line (Figure R3.3j).

Next, we compared the performance of m6Anet models on the Arabidopsis VIRC cell line, where we also observe a comparable performance between the models trained on the human cell lines and the model trained on the Arabidopsis VIRC cell line (Figure R3.4a,b). As an additional comparison, we further include an additional model trained on the HEK293T cell line with labels from miCLIP, m6ACE, and also xPore to match the training data type used in Arabidopsis. Using this model we observe an overall improvement in terms of the ROC AUC and PR AUC on the VIRC dataset (Fig R3.4a,b). Similar to the human data, predictions from all models show a strong enrichment towards the 3'UTR, further supporting that the predicted sites are indeed methylated m6A sites (Figure R3.4c). Lastly, the ranking of k-mer frequencies predicted by m6Anet shows an agreement with the frequencies identified by the original authors (Parker et al) and by xPore, especially among the top ranked motifs (Figure R3.4d,e,f). Most notably, the GGACU motif that is dominant in the original training data is ranked much lower in this dataset, while assigning an accurate ranking towards other more frequently methylated motifs such as AAACU, AGACU, and AGACA that are not as frequently methylated in the original dataset used for training.

Together, these results demonstrate that m6Anet can generalise to datasets with different 5-mer methylated profiles such as direct RNA-Seq from other species.. We thank Reviewer #3 for their suggestions which we believe indeed provides a much better picture on the ability of m6anet to generalise to new data sets. We have included these results in the revised manuscript p. 8 paragraph 2, p. 9 paragraph 1 and in Figure 2, Suppl. Figure 2, Suppl. Figure 3d

Figure R3.3 Comparison of various m6Anet-based models on the HEK293T Test Set
 (a) ROC AUC (b) PR AUC of several m6Anet models on the HEK293T Test Set. m6Anet is the original m6Anet model trained on the HCT116 cell line while Oversampled (Undersampled) refers to m6Anet models trained by oversampling (undersampling) each modified (unmodified) kmer to match the number of unmodified (modified) kmer. 0-neighbor and 1-neighbor kmer are RandomForest model trained on just the one-hot kmer

features surrounding the candidate site.

Figure R3.4 Comparison of m6Anet models on Arabidopsis Datasets

(a-b) ROC Curve and PR Curve of four m6Anet models trained on HCT116 cell line, HEK293T cell line, Arabidopsis VIR-1 complemented cell line, and HEK293T cell line with the inclusion of KO sensitive positions as detected by xPore on the Arabidopsis VIR-1 complemented cell line test set (c) Metagene plot of predicted sites by all four m6Anet models (d) Scatter plot comparing the frequency ranking of predicted motifs by m6Anet against Parker et al and (e) xPore and (f) xPore against Parker et al

Reviewer #3 (point 5):

- m6Anet relies heavily on nanopish to provide event level data (mean, std and dwell) for each transcriptomic position. This should be mentioned and cited in the results section “Training data for m6Anet model parameter estimation”.

Response:

We have followed the suggestion from Reviewer #3 by mentioning Nanopolish and

citing Nanopolish in the results section “Training data for m6Anet model parameter estimation”. Here we present the updated section from the manuscript

Training data for m6Anet model parameter estimation, p.6:

“To learn the model parameters, m6Anet requires training data consisting of labels (modified/unmodified) and direct RNA-Seq reads. In order to train a model for m6A we used labels obtained from m6ACE-Seq that identifies m6A at single nucleotide resolution⁸. m6Anet uses positions which are identified to have m6A as labels for the modified class, and any other position with the same 5-mer sequences that are included in the modified class will be used as the unmodified class. In order to extract features for model training and predictions, we used nanopolish⁹ eventalign to segment nanopore raw signals into their respective positions in the transcriptome. Since m6A modifications occur at the DRACH motifs, we removed any non DRACH motifs from these data for m6Anet, however, this step is not required for training data without prior knowledge about the motifs.[...]”

Reviewer #3:

- Font size on figures is very small throughout, and should be increased.

Response:

Following the suggestion from Reviewer #3, we have updated the font size on figures throughout the manuscript.

Reviewer #3:

- Figure 1c, error bars are hard to read (some are also missing or misaligned) and are not described in the figure legend. I think they have been generated with seaborn barplot so are probably bootstrapped 95% confidence intervals (of per gene kmer mod rate?). Legend also states it is only top 4 kmers but all are shown.

Response:

We have updated Figure 1c so that the error bars are a bit thicker and easier to read. Furthermore, we have also included additional details to the figure legend regarding the confidence intervals of the error bars and also changed top 4 kmers to 18 kmers.

Reviewer #3:

- Fig 2d, m6ACE-seq is labelled in legend as orange but there is no orange line. Supp fig 2b, background is labelled in legend as grey but there is no grey line.

Response:

We thank Reviewer #3 for spotting our errors which we have corrected in the main text. Due to the changes in the manuscript, Fig. 2d is now Fig. 3d in the main text and Suppl. Figure 2b is now Suppl. Figure 3b

Reviewer #3:

- Figure 3e histogram bins are not aligned.

Response:

We have fixed the histogram in Figure 3e and 3f so that the bins are aligned by choosing a fixed bin width of 0.1. Due to the changes in the manuscript, Fig. 3e is now Fig. 2f in the main text.

Reviewer #3:

References:

- Baumgarten S, Bryant JM, Sinha A, Reyser T, Preiser PR, Dedon PC, Scherf A. 2019. Transcriptome-wide dynamics of extensive m6A mRNA methylation during Plasmodium falciparum blood-stage development. *Nat Microbiol* 4:2246–2259.
- Farhat DC, Bowler MW, Communie G, Pontier D, Belmudes L, Mas C, Corrao C, Couté Y, Bougdour A, Lagrange T, Hakimi M-A, Swale C. 2021. A plant-like mechanism coupling m6A reading to polyadenylation safeguards transcriptome integrity and developmental gene partitioning in Toxoplasma. *Elife* 10. doi:10.7554/eLife.68312
- Lee VV, Judd LM, Jex AR, Holt KE, Tonkin CJ, Ralph SA. 2021. Direct Nanopore Sequencing of mRNA Reveals Landscape of Transcript Isoforms in Apicomplexan Parasites. *mSystems* 6. doi:10.1128/mSystems.01081-20
- Mendel M, Delaney K, Pandey RR, Chen K-M, Wenda JM, Vågbo CB, Steiner FA, Homolka D, Pillai RS. 2021. Splice site m6A methylation prevents binding of U2AF35 to inhibit RNA splicing. *Cell* 184:3125-3142.e25.

- Parker MT, Knop K, Sherwood AV, Schurch NJ, Mackinnon K, Gould PD, Hall AJ, Barton GJ, Simpson GG. 2020. Nanopore direct RNA sequencing maps the complexity of Arabidopsis mRNA processing and m6A modification. *Elife* 9. doi:10.7554/eLife.49658
- Roach NP, Sadowski N, Alessi AF, Timp W, Taylor J, Kim JK. 2020. The full-length transcriptome of *C. elegans* using direct RNA sequencing. *Genome Res* 30:299–312.
- Sessegolo C, Cruaud C, Da Silva C, Cologne A, Dubarry M, Derrien T, Lacroix V, Aury J-M. 2019. Transcriptome profiling of mouse samples using nanopore sequencing of cDNA and RNA molecules. *Sci Rep* 9:14908.

Reviewer #4:

Remarks to the Author:

Oxford Nanopore Technologies Nanopore sequencing platform remains the only commercially available sequencing platform that directly measures single RNA and DNA molecules. Hence, it can provide information on the base sequence of DNA and RNA molecules and measure distinct chemical modifications of individual bases of said nucleotide sequences.

While the raw nanopore signal is rich in information, reliably extracting specific parameters such as modification status of RNA bases remains a yet not fully resolved challenge in the field of machine learning. More conventional machine learning methods such as Hidden Markov Models for base and base modification calls have now been replaced by applying deep neural network models. This methodological approach critically relies on extensive training datasets with a known "ground truth". Unfortunately, such datasets currently cannot be generated for all base modifications of interest with the precision required for most deep learning methods, neither with biological nor molecular protocols.

Hendra, Göke, and colleagues address this relevant challenge in their manuscript "Detection of m6A from direct RNA sequencing using a Multiple Instance Learning (MIL) framework by implementing MIL for calling 6mA-modification in nanopore direct RNA sequencing datasets. The authors provide several rationally designed experiments and analyses corroborating the assumption that the performance of their novel approach surpasses those of existing tools and may be a first-in-class tool enabling RNA 6mA modification calling on the single-molecule level using the MIL approach.

Overall, we find m6ANet to be a promising tool for detecting RNAs methylated at 6A using native nanopore sequencing and will recommend it for publication. The use of

the Multiple Instance Learning model is novel and scientifically sound, and it is exciting to see more use of a more comprehensive array of neural network methods in computational biology.

Additionally, the manuscript includes several convincing analyses that their results are concordant with orthogonal (experimental) approaches. Finally, we were easily able to reproduce the results from the paper on CodeOcean as well as easily install the tool on our own servers and run it on the sample data provided in the documentation.

Response:

We thank Reviewer #4 for the positive comments about our manuscript and we are glad that m6Anet can be easily installed and run. We particularly would like to thank Reviewer #4 for their comment on providing a useful estimate of m6Anet precision that directly applies to end users. In our original manuscript we have not provided any such estimate which could easily lead to users performing analysis with sub-optimal parameter settings. We have since then addressed this point by providing precision estimates for specific thresholds for the site level modification probability. Furthermore, we now recommend a default threshold of 0.9 to select m6A sites in the online documentation and we updated our manuscript by explicitly stating the expected precision level that users can expect for a given threshold. We hope that these changes will make the results more accessible and applicable to possible users of m6anet. Please find our detailed response below.

Reviewer #4:

Our criticisms focus mainly on the section entitled Novel m6Anet predictions are sensitive to METTL3 knockout. In the first paragraph, the authors argue that the novel methylation sites predicted by m6ANet are often separately supported by other methylation detection methods and are therefore likely truly methylated. This is not convincing logic as a tool with a very high false-positive rate will, of course, have overlaps with false positives of other methods.

It should also be further clarified that xPore calls differentially methylated sites and is not a "comparative" method to m6ANet. Overall, the take-home message of this paragraph is unclear and ambiguous. A user of your tool would like to know what percentage of the calls made by m6ANet are true positives. However, the authors only explain that m6ANet can reliably detect 46% of KO-sensitive methylation sites. Are these 1888 sites used only novel sites predicted by m6ANet? Can you give a new estimate of the precision of m6ANet if these novel sites are 46% true positives?

Response:

In our original manuscript, we benchmark the performance of m6Anet against other models by comparing their ROC AUC and PR AUC values. We claim that, even though our model outperforms existing methods, the performance of m6Anet is underestimated when we only look at the AUC values derived from miCLIP or m6ACE-seq labels. Reviewer #4 suggests that these results, which are summarised in the paragraph “Novel m6Anet predictions are sensitive to METTL3 knockout”, could be presented more clearly. Specifically, Reviewer #4 has the following comments:

- 1) First paragraph/ Figure 2a: An overlap of predictions from different technologies does not indicate a high precision

This paragraph aims to summarise how the different m6A detection technologies (m6Anet, miCLIP, m6ACe-Seq) overlap in their m6A site predictions. We agree with Reviewer #4 that a high overlap does not indicate if these positions are truly methylated. We have now rephrased this paragraph and combined it with the following paragraph that uses an independent validation set to estimate the fraction of truly methylated sites among the three different protocols.

- 2) Clarification about xPore in relation to m6Anet

We thank Reviewer #4 for highlighting that the description of xPore could be misunderstood. In the revised manuscript we now describe xPore a method that compares different samples (not as a method that is comparative to m6Anet)

- 3) The overall take home message was unclear and ambiguous

In our original manuscript we aimed to demonstrate two points in this paragraph: firstly, we show that technology-specific m6A site predictions are frequently observed and that they can still be true methylated sites, secondly we show that the precision of m6Anet is likely underestimated due to these technology-specific sites. In order to more clearly present these results, we have now split this section into 2 sections (“Technology-specific m6A site predictions are sensitive to METTL3 knockout”, and “m6Anet achieves high precision among top predicted sites”).

- 4) Obtain specific estimates for precision of m6Anet

In our original manuscript we relied on the AUC of the ROC and PR curves to estimate the accuracy of m6Anet predictions in comparison to other methods. These metrics consider the full spectrum of predictions independently from a specific threshold as

they evaluate the ability to rank true positives higher than false positives. m6Anet does not use a default threshold, instead we recommend in our online documentation (<https://m6anet.readthedocs.io/en/latest/>) to use the predicted site probability for ranking candidate sites, therefore we have not provided a point estimate for precision of m6Anet site predictions in our original manuscript. However, we fully agree with Reviewer #4 that such an estimate will be very helpful to make these results interpretable to the reader and to help guide the selection of a meaningful threshold by users. Following this comment, we have therefore estimated the precision of m6Anet site level predictions for different thresholds (Figure R4.1a). At a threshold of 0.9 m6Anet achieved a precision of 70.5%, which we now explicitly mention in the manuscript.

Due to the limitations in the evaluation procedure described above, the precision might be underestimated, even when the additional positions from xPore are considered. In order to obtain accurate estimates of precision we now include an additional analysis on synthetic sequences (“curlcake sequences”) where the labels are known (Liu et al. 2021) (see response to Reviewer #2 for additional details). These data consist of two replicates of a m6A modified library and two replicates of an unmodified IVT RNA library. To obtain precision estimates for different modification rates, we followed the strategy from the authors to randomly sample reads from the modified and unmodified libraries at specific ratios and to exclude 5-mers with multiple modified A nucleotides (Liu et al. 2021). Using the synthetic data, we find that m6Anet achieves very high precision (Figure R4.1b,c). Applying a threshold of 0.9 achieves perfect classification results even with modification rates below 25% (Figure R.4.1d), with lower thresholds showing reduced precision in particular for the lowly modified scenario (Figure R4.1d).

We thank Reviewer #4 for their suggestion. We hope that the additional validation on the synthetic dataset, and the detailed analysis of precision for different thresholds in m6Anet will provide a more comprehensive and clearer guide on how to interpret the results from m6Anet.

Figure R4.1 Comparison of m6Anet models on Arabidopsis Datasets

(a) The adjusted precision true positive rate after including position sensitive to METTL3-KO of m6Anet and EpiNano. Red dots indicate the precision point at different m6Anet output thresholds (b-c) Boxplots comparing the ROC AUC (b) and PR AUC (c) of m6Anet on curlcake datasets over different mixtures of methylated reads. (d) Precision of m6Anet on the curlcake datasets for varying methylation level at different predictive thresholds.

Reviewer #4:

Other minor criticisms include:

- (Supp) Figure 2c has "Precision" as the y axis label but then describes the true positive rate in the legend. This is done twice
- Figure 2b has a

mismatched color legend

Response:

We thank Reviewer #4 for finding this error in Figure 2b, Figure 2c, and Supp Figure 2c. We have corrected the figure legends to match the colours of the plotted lines and corrected the y axis label and figure legends to describe precision instead of true positive rate in the revised manuscript.

1. Parker, M. T. *et al.* Nanopore direct RNA sequencing maps the complexity of Arabidopsis mRNA processing and m6A modification. *Elife* **9**, (2020).
2. Garalde, D. R. *et al.* Highly parallel direct RNA sequencing on an array of nanopores.

Nat. Methods **15**, 201–206 (2018).

3. Chen, Y. *et al.* A systematic benchmark of Nanopore long read RNA sequencing for transcript level analysis in human cell lines. doi:10.1101/2021.04.21.440736.
4. Aw, J. G. A. *et al.* Determination of isoform-specific RNA structure with nanopore long reads. *Nat. Biotechnol.* **39**, 336–346 (2021).
5. Stephenson, W. *et al.* Direct detection of RNA modifications and structure using single molecule nanopore sequencing. *bioRxiv* 2020.05.31.126763 (2020) doi:10.1101/2020.05.31.126763.
6. Liu, H. *et al.* Accurate detection of m6A RNA modifications in native RNA sequences.

Nat. Commun. **10**, 4079 (2019).

7. Zhang, T. *et al.* RNALocate: a resource for RNA subcellular localizations. *Nucleic Acids Res.* **45**, D135–D138 (2017).
8. Koh, C. W. Q., Goh, Y. T. & Sho Goh, W. S. Atlas of quantitative single-base-resolution N6-methyl-adenine methylomes. *Nature Communications* vol. 10 (2019).

9. Loman, N. J., Quick, J. & Simpson, J. T. A complete bacterial genome assembled de novo using only nanopore sequencing data. *Nat. Methods* **12**, 733–735 (2015).

Decision Letter, first revision:

Subject: AIP Decision on Manuscript NMETH-A47051B
Message: Our ref: NMETH-A47051B

7th Jul 2022

Dear Jonathan,

Thank you for submitting your revised manuscript "Detection of m6A from direct RNA sequencing using a Multiple Instance Learning framework" (NMETH-A47051B). It has now been seen by the original referees and their comments are below. The reviewers find that the paper has improved in revision, and therefore we'll be happy in principle to publish it in Nature Methods, pending minor revisions to satisfy the referees' final requests and to comply with our editorial and formatting guidelines.

TRANSPARENT PEER REVIEW

Nature Methods offers a transparent peer review option for new original research manuscripts submitted from 17th February 2021. We encourage increased transparency in peer review by publishing the reviewer comments, author rebuttal letters and editorial decision letters if the authors agree. Such peer review material is made available as a supplementary peer review file. Please state in the cover letter 'I wish to participate in transparent peer review' if you want to opt in, or 'I do not wish to participate in transparent peer review' if you don't. Failure to state your preference will result in delays in accepting your manuscript for publication.

Thank you again for your interest in Nature Methods Please do not hesitate to contact me if you have any questions.

Best regards,
Lei

Lei Tang, Ph.D.
Senior Editor
Nature Methods

ORCID

Reviewer #1 (Remarks to the Author):

The authors have satisfied all of my concerns, especially with the detailed analysis of the Arabidopsis data from Parker et al.

Reviewer #2 (Remarks to the Author):

The authors have updated the density plots, the Venn diagrams, the code base, and the online documentation well, and the cross-species validation data (especially the titration data) look good. They have satisfactorily addressed all my concerns.

Reviewer #3 (Remarks to the Author):

Overall, I would like to commend the authors for their extremely thorough and impressive response to the reviewers comments, which were meant constructively! I recommend that the paper should be accepted for publication, with a few further clarifications to the text.

The authors have performed a very thorough benchmarking of different over- and under-sampling strategies to evaluate their dataset. I am not sure that I agree with their interpretation of the data, however. It is clear that over- or under-sampling of kmers, so that positive and negative training sets have similar distributions, has a negative impact on performance when evaluated on the species that the training data is derived from (e.g. models trained on HEK293T evaluated on HEK293T or HCT116, or models trained on Arabidopsis evaluated on Arabidopsis). This is to be expected as it prevents m6Anet

from learning/overfitting to the kmer distribution of the organism used for training. In multiple cases however, over or under- sampling does improve generalisation when applied to different organisms: e.g. models trained on HEK data were more predictive on Arabidopsis when undersampling was used, and models trained on HCT data were more predictive on Arabidopsis when either over- or under-sampling was used. Some of these changes in AUC are small and are probably within margins of error, which are not estimated. However, I think the clearest indicator of species-specific differences is that models trained on human cell lines perform more poorly on Arabidopsis (AUCs around 0.88) than models trained on Arabidopsis data (AUCs around 0.94), and vice-versa.

I think the authors can address this issue by changing the text. It is not reasonable to state, as the authors do on line 230 of the revised manuscript, that “m6Anet generalises robustly to other cell lines and species without a loss in accuracy due to cell type-specific training data”. The AUCs show clearly that there is a loss in accuracy due to species-specific training data. Rather than claiming that a single model trained on human data provides a “one-size-fits-all” solution that can be applied to any dataset or organism, they should recognise that different model weights should be used for different circumstances. Where a model or training dataset exists for a specific organism, users could use or train a species-specific model that has learned the kmer biases of that organism. When analysing a species for which it is not currently possible to train a model, then models which have been trained on closely related organisms, or more distantly related organisms but with balanced training data to remove kmer bias, might be preferable. Similar solutions have been recommended for nanopore basecalling models, where species-specific differences in kmer and/or modification content affect the performance of basecalling when models trained in one species are applied to other species (Wick et al., 2019).

My only other comment is that the Arabidopsis datasets are not generated from cell lines, but from whole organisms. The authors should replace references in the text to “Arabidopsis cell line” with “Arabidopsis accession” when referring to the genetic background (Colombia-0, Col-0), and “Arabidopsis mutant” or “Arabidopsis line” when referring to mutants (vir-1) or transgenics (VIRc) respectively.

Wick RR, Judd LM, Holt KE. 2019. Performance of neural network basecalling tools for Oxford Nanopore sequencing. *Genome Biol* 20:129.

Reviewer #4 (Remarks to the Author):

We have reviewed the revised manuscript “Detection of m6A from direct RNA sequencing using a Multiple Instance Learning framework” by Hendra, Göke and colleagues.

Our previous concerns centered around the section entitled "Novel m6Anet predictions are sensitive to METTL3 knockout". The authors have answered our specific request for a clearer estimate of the resulting precision by leveraging the results of experiments requested by other reviewers. ie. analysis of

other species genomes and synthetic RNA molecules in the new section entitled "m6Anet achieves high precision among top predicted sites".

Furthermore, the authors also demonstrated in their revision that the neural network is extendable to other species and appears not to be overfit on human data. The use of synthetic RNA with known modification sites as well as known stoichiometry provides convincing evidence for the benefit of this tool.

We note that the authors have added further work with existing tools (eg nanoRMS) despite none of the existing tools being based on neural networks. The comparative performance to m6Anet further emphasizes our previous impression that m6Anet is a promising application of deep learning in genomics and should help pave the way for further method development and optimization.

Given that already the initial manuscript described an novel, interesting and potentially impactful application of machine learning methods on nanopore sequencing data and the authors have addressed our concerns as well as - from our perspective - those of the other referees satisfactorily, we recommend the manuscript for publication.

Author Rebuttal, first revision:

We would like to thank all Reviewers for their positive and helpful comments during the revision process. We have made the remaining minor changes as suggested (see below for a point-by-point response). In addition, we have made several minor edits to shorten the manuscript in response to the editorial request. We also corrected one sentence in the Methods, which was accidentally changed during formatting of the last submission.

Reviewer #1 (Remarks to the Author):

The authors have satisfied all of my concerns, especially with the detailed analysis of the Arabidopsis data from Parker et al.

Reviewer #2 (Remarks to the Author):

The authors have updated the density plots, the Venn diagrams, the code base, and the online documentation well, and the cross-species validation data (especially the titration data) look good. They have satisfactorily addressed all my concerns.

Reviewer #3 (Remarks to the Author):

Overall, I would like to commend the authors for their extremely thorough and impressive response to the reviewers comments, which were meant constructively! I recommend that the paper should be accepted for publication, with a few further clarifications to the text.

The authors have performed a very thorough benchmarking of different over- and under-sampling strategies to evaluate their dataset. I am not sure that I agree with their interpretation of the data, however. It is clear that over- or under-sampling of kmers, so that positive and negative training sets have similar distributions, has a negative impact on performance when evaluated on the species that the training data is derived from (e.g. models trained on HEK293T evaluated on HEK293T or HCT116, or models trained on Arabidopsis evaluated on Arabidopsis). This is to be expected as it prevents m6Anet from learning/overfitting to the kmer distribution of the organism used for training. In multiple cases however, over or under- sampling does improve generalisation when applied to different organisms: e.g. models trained on HEK data were more predictive on Arabidopsis when undersampling was used, and models trained on HCT data were more predictive on Arabidopsis when either over- or under-sampling was used. Some of these changes in AUC are small and are probably within margins of error, which are not estimated. However, I think the clearest indicator of species-specific differences is that models trained on human cell lines perform more poorly on Arabidopsis (AUCs around 0.88) than models trained on Arabidopsis data (AUCs around 0.94), and vice-versa.

I think the authors can address this issue by changing the text. It is not reasonable to state, as the authors do on line 230 of the revised manuscript, that “m6Anet generalises robustly to other cell lines and species without a loss in accuracy due to cell type-specific training data”. The AUCs show clearly that there is a loss in accuracy due to species-specific training data. Rather than claiming that a single model trained on human data provides a “one-size-fits-all” solution that can be applied to any dataset or organism, they should recognise that different model weights should be used for different circumstances. Where a model or training dataset exists for a specific organism, users could use or train a species-specific model that has learned the kmer biases of that organism. When analysing a species for which it is not currently possible to train a model, then models which have been trained on closely related organisms, or more distantly related organisms but with balanced training data to remove kmer bias, might be preferable. Similar solutions have been recommended for nanopore basecalling models, where species-specific differences in kmer and/or modification content affect the performance of basecalling when models trained in one species are applied to other species (Wick et al., 2019).

Response:

We thank Reviewer #3 for highlighting this point. We have now changed the sentence and claim accordingly.

“These data demonstrate that *m6Anet* generalises robustly to other cell lines without a loss in accuracy due to cell type-specific data. While a species-specific model will provide best results, in the absence of a species-specific training data, *m6Anet* still provides accurate predictions even when the default human-trained model is used.”

Reviewer #3:

My only other comment is that the Arabidopsis datasets are not generated from cell lines, but from whole organisms. The authors should replace references in the text to “Arabidopsis cell line” with “Arabidopsis accession” when referring to the genetic background (Colombia-0, Col-0), and “Arabidopsis mutant” or “Arabidopsis line” when referring to mutants (*vir-1*) or transgenics (*VIRc*) respectively.

Response:

We have now changed the text accordingly as suggested.

Wick RR, Judd LM, Holt KE. 2019. Performance of neural network basecalling tools for Oxford Nanopore sequencing. *Genome Biol* 20:129.

Reviewer #4 (Remarks to the Author):

We have reviewed the revised manuscript “Detection of m6A from direct RNA sequencing using a Multiple Instance Learning framework” by Hendra, Göke and colleagues.

Our previous concerns centered around the section entitled “Novel *m6Anet* predictions are sensitive to METTL3 knockout”. The authors have answered our specific request for a clearer estimate of the resulting precision by leveraging the results of experiments requested by other reviewers. ie. analysis of other species genomes and synthetic RNA molecules in the new section entitled “*m6Anet* achieves high precision among top predicted sites”.

Furthermore, the authors also demonstrated in their revision that the neural network is extendable to other species and appears not to be overfit on human data. The use of synthetic RNA with known modification sites as well as known stoichiometry provides convincing evidence for the benefit of this tool.

We note that the authors have added further work with existing tools (eg nanoRMS) despite none of the existing tools being based on neural networks. The comparative performance to *m6Anet* further emphasizes our previous impression that *m6Anet* is a promising application of deep learning in genomics and should help pave the way for further method development and

optimization.

Given that already the initial manuscript described an novel, interesting and potentially impactful application of machine learning methods on nanopore sequencing data and the authors have addressed our concerns as well as - from our perspective - those of the other referees satisfactorily, we recommend the manuscript for publication.

Final Decision Letter:

27th Sep 2022

Dear Dr Goeke,

I am pleased to inform you that your Article, "Detection of m6A from direct RNA sequencing using a Multiple Instance Learning framework", has now been accepted for publication in Nature Methods. Your paper is tentatively scheduled for publication in our December print issue, and will be published online prior to that. The received and accepted dates will be 7th Sep 2021 and 27th Sep 2022. This note is intended to let you know what to expect from us over the next month or so, and to let you know where to address any further questions.

Over the next few weeks, your paper will be copyedited to ensure that it conforms to Nature Methods style. Once your paper is typeset, you will receive an email with a link to choose the appropriate publishing options for your paper and our Author Services team will be in touch regarding any additional information that may be required.

Your paper will now be copyedited to ensure that it conforms to Nature Methods style. Once proofs are generated, they will be sent to you electronically and you will be asked to send a corrected version within 24 hours. It is extremely important that you let us know now whether you will be difficult to contact over the next month. If this is the case, we ask that you send us the contact information (email, phone and fax) of someone who will be able to check the proofs and deal with any last-minute problems.

If, when you receive your proof, you cannot meet the deadline, please inform us at rjsproduction@springernature.com immediately.

Once your manuscript is typeset and you have completed the appropriate grant of rights, you will receive a link to your electronic proof via email with a request to make any corrections within 48 hours. If, when you receive your proof, you cannot meet this deadline, please inform us at rjsproduction@springernature.com immediately.

Once your paper has been scheduled for online publication, the Nature press office will be in touch to confirm the details.

Content is published online weekly on Mondays and Thursdays, and the embargo is set at 16:00 London time (GMT)/11:00 am US Eastern time (EST) on the day of publication. If you need to know the exact publication date or when the news embargo will be lifted, please contact our press office after you have submitted your proof corrections. Now is the time to inform your Public Relations or Press Office about your paper, as they might be interested in promoting its publication. This will allow them time to prepare an accurate and satisfactory press release. Include your manuscript tracking number NMETH-A47051C and the name of the journal, which they will need when they contact our office.

About one week before your paper is published online, we shall be distributing a press release to news organizations worldwide, which may include details of your work. We are happy for your institution or funding agency to prepare its own press release, but it must mention the embargo date and Nature Methods. Our Press Office will contact you closer to the time of publication, but if you or your Press Office have any inquiries in the meantime, please contact press@nature.com.

Please note that Nature Methods is a Transformative Journal (TJ). Authors may publish their research with us through the traditional subscription access route or make their paper immediately open access through payment of an article-processing charge (APC). Authors will not be required to make a final decision about access to their article until it has been accepted. Find out more about Transformative Journals

Authors may need to take specific actions to achieve compliance with funder and institutional open access mandates. If your research is supported by a funder that requires immediate open access (e.g. according to Plan S principles) then you should select the gold OA route, and we will direct you to the compliant route where possible. For authors selecting the subscription publication route, the journal's standard licensing terms will need to be accepted, including self-archiving policies. Those licensing terms will supersede any other terms that the author or any third party may assert apply to any version of the manuscript.

To assist our authors in disseminating their research to the broader community, our SharedIt initiative provides you with a unique shareable link that will allow anyone (with or without a subscription) to read the published article. Recipients of the link with a subscription will also be able to download and print the PDF. As soon as your article is published, you will receive an automated email with your shareable link.

Please note that you and your coauthors may order reprints and single copies of the issue containing your article through Nature Portfolio 's reprint website, which is located at <http://www.nature.com/reprints/author-reprints.html>. If there are any questions about reprints please send an email to author-reprints@nature.com and someone will assist you.

Best regards,
Lei

Lei Tang, Ph.D.
(she/her/hers)
Senior Editor
Nature Methods
Follow us @naturemethods @DrLeiTang

** Visit the Springer Nature Editorial and Publishing website at www.springernature.com/editorial-and-publishing-jobs for more information about our career opportunities. If you have any questions please click here.**

Research Square's research promotion services help your work get seen and understood by the people you want to reach most. With Video Bytes, Custom Infographics, Visual Abstracts, and more, make your work stand out and increase the effectiveness of your outreach.

Learn more and get started now!

This email has been sent through the Springer Nature Tracking System NY-610A-NPG&MTS

Confidentiality Statement:

This e-mail is confidential and subject to copyright. Any unauthorised use or disclosure of its contents is prohibited. If you have received this email in error please notify our Manuscript Tracking System Helpdesk team at <http://platformsupport.nature.com> .

Details of the confidentiality and pre-publicity policy may be found here

<http://www.nature.com/authors/policies/confidentiality.html>

Privacy Policy | Update Profile